# Integrating multi-omics and machine learning to decipher the molecular pathways of bisphenol a-associated lactylation-related genes driving bladder cancer

Hao Wang[1©], Hongquan Liu[2©], Fengze Sun[2], Jitao Wu[2]*

**1** School of Clinical Medicine, Shandong Second Medical University, Weifang, Shandong, People's Republic of China, **2** Department of Urology, The Affiliated Yantai Yuhuangding Hospital of Qingdao University, Yantai, Shandong, People's Republic of China

☯ These authors contributed equally to this work
* wjturology@163.com

## Abstract

In this study, we systematically investigated bladder cancer–related gene signatures using a toxicogenomics-informed framework, with particular attention to genes associated with lactylation-related pathways. Multi-omics data from the Gene Expression Omnibus (GEO) and The Cancer Genome Atlas (TCGA) were integrated, and Weighted Gene Co-expression Network Analysis (WGCNA), a toxicology database, and lactylation-related gene sets were combined for intersection screening. Machine learning algorithms, including LASSO, SVM, and random forest, were then applied to identify key genes. Four prioritized BPA–lactylation-associated candidate genes—ENO1, WBP11, GTF2F1, and SPR—were ultimately identified and showed consistent associations with metabolic, immune, and transcription-related features. Multi-level validation, including immune infiltration analysis, single-cell transcriptome localization, proteomic validation, and molecular docking and kinetic simulation, supported the structural plausibility of BPA–protein interactions at the molecular level. This study proposes a toxicogenomics-informed, hypothesis-generating framework that prioritizes candidate genes and pathways potentially linking BPA-related signatures with lactylation-associated processes in bladder cancer.

## Introduction

Bladder cancer is one of the most common malignant tumors of the urinary system worldwide and imposes a substantial burden on patients and healthcare systems. Its pathogenesis is influenced by environmental, genetic, and epigenetic factors [1]. In recent years, environmental toxicants have been increasingly recognized as important exogenous contributors to bladder carcinogenesis.[2–4]. As a common endocrine disruptor, BPA is widely used in the production of synthetic polymers and thermal paper

**Data availability statement:** "Yes - all data are fully available without restriction; All public datasets used in this study are available via official access: GEO Datasets GSE13507: https://www.ncbi.nlm.nih.gov/geo/query/acc.cgi?acc=GSE13507; Search term: bladder cancer GSE13507 GSE130001: https://www.ncbi.nlm.nih.gov/geo/query/acc.cgi?acc=GSE130001; Search term: bladder cancer single-cell GSE130001 GSE145281: https://www.ncbi.nlm.nih.gov/geo/query/acc.cgi?acc=GSE145281; Search term: bladder cancer immunotherapy single-cell GSE145281 External validation cohorts: GSE154261, GSE69795, GSE31684, GSE19423, GSE39281 TCGA Dataset TCGA-BLCA: https://portal.gdc.cancer.gov/projects/TCGA-BLCA; Search term: TCGA bladder cancer BLCA".

**Funding:** This work was supported by the National Natural Science Foundation of China (No. 82370690), Natural Science Foundation of Shandong Province (No. ZR2023MH241), basic research project of Yantai Science and Technology Innovation development plan (No.2023JCYJ069) and Shandong Health Science Innovation Team Building Project. The funders had no role in study design, data collection and analysis, decision to publish, or preparation of the manuscript.

**Competing interests:** The authors have declared that no competing interests exist.

and can accumulate in the human body [5]. BPA has been shown to exert pro-tumorigenic effects in various cancers through the activation of estrogen receptors, promotion of oxidative stress, and metabolic disruption [6,7]. Although the effects of BPA have been well studied in breast cancer, its mechanism of action in bladder cancer needs to be further explored [8]. However, there is still limited evidence regarding whether and how BPA may be involved in bladder cancer pathogenesis at a systems or mechanistic level.

Epigenetic modifications play an important role in the occurrence and development of bladder cancer. Recent studies have shown that multiple epigenetic mechanisms, including DNA methylation, histone modifications, and non-coding RNAs, are involved in bladder cancer progression. For example, changes in DNA methylation can lead to aberrant gene expression, thereby affecting tumor growth and metastasis [9]. In addition, epigenetic alterations have been linked to chemoresistance in bladder cancer and may offer potential targets for personalized therapy through the regulation of key genes [10]. Lactylation, a newly discovered post-translational modification, also plays an important role in the tumor microenvironment. Lactate is a major product of glycolysis, while lactylation is a protein modification induced by lactate [11]. Studies have shown that lactylation can be involved in tumor development through downstream transcriptional regulation and is a potential target for tumor therapy [12]. In bladder cancer, lactate modification may promote tumor progression and metastasis by affecting metabolic reprogramming and epigenetic regulation of tumor cells [13]. Although previous studies have suggested the functional potential of lactylation in bladder tumors, its specific role in bladder cancer remains unclear, and systematic investigations into its potential links with environmental toxicants are still lacking.

Therefore, the aim of this study was to systematically identify and prioritize potential BPA-associated, lactylation-related candidate genes involved in bladder cancer using integrative bioinformatics and multi-level validation. Through a systematic multi-omics integration strategy, we constructed an analysis framework based on multi-platform datasets, including GEO and TCGA. WGCNA, a toxicology database, and a lactylation-related gene set were combined for intersection screening, and machine learning algorithms, including Random Forest and LASSO, were used to identify core targets. These candidates were further evaluated through immune infiltration analysis, single-cell sequencing, proteomic validation, and molecular docking. Four key BPA-associated lactylation-related genes—ENO1, WBP11, GTF2F1, and SPR—were identified, suggesting that BPA-related signatures may be linked to the lactylation network and to metabolic, immune, and transcriptional features of bladder cancer. This study not only provides a new epigenetic perspective to elucidate the mechanism of BPA-associated bladder cancer, but also provides a theoretical basis and potential targets for the search for biomarkers and targeted interventions for environmental exposure-associated tumors.

## Methods

### Data sources and preprocessing

Transcriptomic and single-cell sequencing data from several publicly available databases were used in this study. The bladder cancer bulk transcriptome expression

data were obtained from the GSE13507 [14,15] dataset in the GEO database, which contains a total of 191 samples, including primary bladder cancer tissues, recurrent tumors, paracancerous tissues, and normal bladder mucosal tissues. To unify the analysis, we combined 58 of the paracancerous tissues with 10 normal bladder mucosa samples into the normal group, and the rest were categorized into the disease group. The data platform was Illumina Human-6 v2.0 expression BeadChip, and the raw expression matrix was log2-transformed and normalized. And to eliminate the non-biological differences caused by the technical sources, the data were corrected for batch effects by applying the ComBat method in the R language 4.3.2#39;s "sva" package. The data were corrected for batch effects by the ComBat method in the "sva" package of R4.3.2, and the distribution of the samples before and after correction was visualized by principal component analysis (PCA) plots, which showed that the data quality met the requirements of the analysis. Toxicity target genes related to BPA were obtained from the Comparative Toxicogenomics Database [16] (CTD, https://ctdbase.org) with the search keyword "bisphenol A", and the screening species was "Homo sapiens"; targets with experimental or literature support were retained for subsequent intersection analysis. The collection of lactylation-related genes, on the other hand, was obtained from previously published literature, covering identified enzymes, substrate proteins, and regulatory factors related to lysine lactylation modification, to establish the set of lactylation genes for analysis. These annotations reflect curated toxicogenomics knowledge and do not represent patient-level BPA exposure measurements. In addition, to further analyze the expression characteristics of key genes in different cell types in the tumor microenvironment, two single-cell RNA sequencing datasets were introduced: GSE130001 [17,18] and GSE145281 [19]. The GSE130001 dataset was derived from bladder cancer tissues, enriched with tumor and stromal cells after CD45-negative screening, and is suitable for the analysis of structural cell taxa; GSE145281 contains peripheral blood mononuclear cells (PBMC) from bladder cancer patients undergoing immunotherapy and can be used to assess the expression profile of key genes in immune cells. Both datasets were downloaded from the GEO database and combined with standardized annotation information provided by the TISCH2 platform for subsequent single-cell level expression analysis [20]. In this study, "lactylation-related genes" refer to genes previously reported to participate in lactylation-associated metabolic or regulatory pathways, rather than proteins directly confirmed to carry lysine lactylation modifications in the analyzed samples.

## WGCNA analysis and intersecting gene screening

Weighted Gene Co-expression Network Analysis (WGCNA) is a scale-free network construction method based on gene expression profiling data, which can identify gene modules with co-expression trends among samples through clustering analysis and correlation analysis with clinical phenotypes to screen potential key regulatory factors [21]. To identify functional modules closely related to bladder cancer and screen potential key genes, this study constructed a weighted gene co-expression network based on the "WGCNA" package (v1.72-1) in R language. Input data is the gene expression matrix in GSE13507. The samples were first analyzed by hierarchical clustering to identify and exclude abnormal samples to ensure the stability of the network construction. Subsequently, the scale-free topology criterion was used to filter the appropriate soft threshold (β value), and the "pickSoftThreshold" function was used to calculate the $R^2$ and the average connectivity index under different powers, and the soft thresholds with $R^2 > 0.85$ and obvious curve inflection points were selected. The "pickSoftThreshold" function is used to calculate the $R^2$ and the average connectivity index under different powers, and select the soft threshold that satisfies the $R^2 > 0.85$ and the curve inflection point is obvious to construct the weighted adjacency matrix, which is further converted into the Topological Overlap Matrix (TOM). Based on the dissimilarity distance of the TOM, the module identification was performed using the dynamic tree cut (dynamic tree cut) method, setting the minimum module gene number to 50, merging the module eigenvectors (eigengenes) of the highly similar modules, and the module merging threshold was set to 0.25. After the construction was completed, the Pearson correlation coefficients between the modules and the phenotypes (bladder cancer) were calculated, and the heat map of module-trait correlation was generated. All module genes were extracted from the significantly correlated modules as candidate gene sets, which were further intersected and analyzed with the set of BPA-related target genes retrieved from the Comparative

Toxicogenomics Database, as well as the set of lactated genes obtained from literature collation. The intersection operation was realized in R4.3.2 using the "VennDiagram" package, and the obtained genes were used for subsequent functional annotation and core gene identification analysis [22]. To evaluate whether the observed overlap among the BPA target genes, lactylation-related genes, and WGCNA module genes exceeded random expectation, statistical calibration analyses were performed. Specifically, random gene sets matched for size to each of the three gene lists were repeatedly sampled from the background gene universe, and the distribution of intersection sizes was estimated under null conditions. This approach was used to assess the potential influence of gene-set size effects and module selection variability on the observed overlap.

## Immune infiltration analysis

To comprehensively assess the infiltration characteristics of immune cells in bladder cancer tissues, this study used three classical immune infiltration algorithms based on the GSE13507 expression matrix in the GEO database: the CIBERSORT, the single-sample Gene Set Enrichment Analysis (ssGSEA), and the MCPcounter, to quantify the immune cell composition in the tumor microenvironment [23–25]. All analyses were performed in R 4.3.2. CIBERSORT analysis was performed by deconvolution of the sample expression data using the "CIBERSORTx" algorithm in conjunction with the LM22 immune cell signature matrix to calculate the relative proportions of the 22 immune cell types in each sample. The number of permutations was set to 1000, and the results were filtered by P-value < 0.05. ssGSEA analysis was implemented using the "GSVA" package (v1.48.3), which scores each sample based on the set of genes related to immune cell function, reflecting the enrichment degree of immune cells among different samples [26]. MCPcounter analysis was performed using the MCPcounter module in the "immunedeconv" package to quantitatively evaluate more than 10 types of immune and mesenchymal cells, including T cells, NK cells, monocytes, endothelial cells, and so on [27]. For the visualization of the results, heatmaps were created using the "ComplexHeatmap" package (v2.18.0), which demonstrated the overall distribution trend of immune cell abundance between the normal and tumor groups calculated by different methods [28]. To further explore the synergistic trend of different immune cell subpopulations in bladder cancer tissues, we calculated the correlation matrices between immune cells based on three algorithms, namely, CIBERSORT, ssGSEA, and MCPcounter, and the correlation analyses were performed by using "corrplot" and "ggcorrplot". Correlation analysis was performed using the "corrplot" and "ggcorrplot" packages to draw intragroup correlation bubble plots, and the graphs were uniformly scaled with a color scale and labeled with statistical significance (P < 0.05) [29].

## Expression characterization and functional network analysis of intersecting genes

To further explore the expression characteristics of BPA-associated lactylation genes in bladder cancer tissues and their potential biological functions, this study first extracted the expression information of the intersecting genes in samples of the tumor group and normal group based on the GSE13507 expression matrix. After the expression matrix was log2-transformed and Z-score normalized, the expression heatmap was plotted using the "pheatmap" package (v1.0.12) in R4.3.2 to visualize the expression differences between the intersected genes in different groups. Subsequently, to analyze the functional interactions among the intersecting genes, we constructed a protein-protein interaction network with the help of the STRING database (Search Tool for the Retrieval of Interacting Genes/Proteins, https://string-db.org) [30]. STRING is a high-quality protein-protein interaction database that integrates experimental data, literature mining, computational prediction, and other sources, and supports multiple species, including human. In the analysis, the species was set as "Homo sapiens", the minimum interaction confidence score threshold was 0.4 (medium confidence), and the results were restricted to direct interactions between intersecting genes. The constructed PPI network was used to identify potential synergistic modules and signaling relationships. We further performed Gene Set Enrichment Analysis (GSEA) using the clusterProfiler package (v4.6.2) in R to identify signaling pathways and biological functions associated with the intersected gene set. The GSEA was conducted based on a pre-ranked list of all genes derived from the differential expression

analysis between tumor and normal tissues, using the signal-to-noise ratio as the ranking metric. Gene sets from the Molecular Signatures Database (MSigDB) Hallmark and KEGG collections were tested. Significance was assessed through 1,000 permutations, and enrichment results with a false discovery rate (FDR) q-value < 0.25 were considered statistically significant. The outcomes were visualized using a dot plot to display enrichment strength (-log10(P value)), gene counts, and the significance of the top-enriched pathways. Finally, to further characterize potential interaction networks involving BPA-related targets and bladder cancer–associated genes, we constructed a "BPA-target-tumor" network using Cytoscape 3.7.2 [31].

## Machine learning algorithms for screening key genes

To accurately identify key genes with potential diagnostic value for bladder cancer from the intersecting genes, this study integrates three classical machine learning methods: LASSO regression (Least Absolute Shrinkage and Selection Operator) [32], Support Vector Machine (SVM) [33], and Random Forest (RF) [34]. All the analyses were done based on the R language (v4.3.2) platform, and the input data were the expressions of the intersected genes in the GSE13507 dataset in the tumor group and the normal group.1) LASSO regression is a linear regression method used to deal with high-dimensional data, which has the dual functions of variable filtering and dimensionality reduction. The analysis was implemented by the "glmnet" package (v4.1-8), using binomial logistic regression mode (family = "binomial"), setting the standardization parameter standardize = TRUE and passing a 10-fold cross-validation (nfolds). 10-fold cross-validation (nfolds = 10) to determine the optimal penalty parameter λ. Finally, genes with non-zero regression coefficients are extracted as candidate features at the corresponding points of lambda. Min. 2) SVM is a feature recursive elimination algorithm combined with a support vector machine classifier, which is capable of identifying the optimal subset of features iteratively. In this study, we use the "e1071" package (v1.7-13) to construct a linear kernel function support vector machine (kernel = "linear", cost = 1), combined with the "caret "package (v1.0-94) to perform feature recursion with 5-fold cross-validation (cv = 5), calculate the cross-validation error and accuracy under different number of features, and finally select the feature with the smallest error and the highest accuracy as the optimal gene set. 3) Random forest is an integrated learning method based on decision trees, which is based on the construction of a large number of tree models and integrating their prediction results to improve model stability. The analysis uses the "randomForest" package (v4.7-1.1), and sets the number of decision trees ntree = 500 and the number of feature variables mtry = sqrt(p), where p is the total number of features. The top 30 genes with the highest contribution values were extracted based on the Mean Decrease Gini (MDG) ranking of each feature in node partitioning. Finally, the feature gene sets obtained by the three algorithms were intersected and integrated by the "VennDiagram" package, and the key genes identified by all three methods were screened out as the final candidate targets.

## Diagnostic model construction and validation

To further evaluate the diagnostic potential of the four screened key genes, multiple machine learning classification models were constructed using the training dataset (TCGA+GTEx) and validated in an independent external cohort (GSE13507). The modeling and visualization were performed in R (v4.3.2), with major packages including "caret", "caretEnsemble", "glmnet", "randomForest", "pROC" and "ggplot2". To minimize potential batch effects between TCGA and GTEx datasets, expression data were processed and normalized within each cohort prior to model training. Feature selection was performed exclusively within the training dataset to avoid information leakage, and the selected features were then applied to the external validation cohort.

We adopted a 5-fold cross-validation strategy to reduce the risk of overfitting. Various algorithms were applied, including generalized linear model boosting (glmBoost), elastic net regression (Enet) with different α parameters, ridge regression, lasso regression, support vector machine (SVM), stepwise regression (Stepglm), and their combinations. For each model, the area under the receiver operating characteristic curve (AUC) was calculated in both training and validation

datasets to evaluate diagnostic performance. ROC curves were plotted to visualize model discrimination ability. Model performance should be interpreted as exploratory and predictive, rather than as definitive diagnostic evidence.

## Validation of expression levels, evaluation of diagnostic efficacy, and prognostic analysis of key genes

To verify the expression pattern of the screened key genes in bladder cancer with diagnostic and prognostic value, expression analysis, ROC analysis, and survival analysis were performed in two independent cohorts, GSE13507 and TCGA-BLCA, respectively. All statistical analyses were done in the R language 4.3.2 environment, and the main R packages used included "limma" (v3.52.2), "ggplot2" (v3.4.4), "pROC "(v1.18.4), "survival" (v3.3-1), and "survminer" (v0.4.9). In the expression level analysis, the "limma" package was used to compare the differences between the tumor group and the normal group samples in GSE13507 and TCGA, and the four key genes were presented by "ggplot2" violin plots (SPR, The expression differences of four key genes (SPR, WBP11, ENO1, GTF2F1) in different tissues were presented in violin plots by "ggplot2", and the P-values were calculated by two-sided t-test. To evaluate the diagnostic efficacy of the key genes, ROC curves were constructed based on the GSE13507 and TCGA expression matrices, and the AUC (Area Under the ROC Curve) was calculated and visualized using the "pROC" package, and the higher the AUC value was, the more accurate it was in distinguishing between tumor and normal samples. The higher the AUC value, the better the accuracy in distinguishing between tumor and normal samples. For prognostic analysis, a total of 411 patients were retained for subsequent analysis after excluding normal tissue samples and samples with missing survival information using the TCGA-BLCA project RNA-seq (STAR process, TPM format) and clinical data, combined with the follow-up data published in Cell (2018) by Liu et al. Expression data were first log2(value+1) transformed, and then the association of the four genes with overall survival (OS, Overall Survival) was assessed based on survival data. The Cox proportional risk model was fitted using the "survival" package, and the optimal cut-off value was calculated using the surv_cutpoint function in the "survminer" package for sample grouping. Survival differences were estimated by the Kaplan-Meier method, and the log-rank test was used to compare the survival differences between the two groups, with the significance level set at P < 0.05. The forest plots and the survival differences were finally calculated by the "ggplot2" and "survminer" packages. Finally, the forest plot and survival curve were plotted by "ggplot2" and "survminer".

## Localization analysis of key genes

To further analyze the cell-type-specific expression of the four key genes in the bladder cancer microenvironment, we introduced two complementary single-cell RNA sequencing datasets from the GEO database, namely GSE130001 and GSE145281.

GSE130001 was derived from bladder cancer tissues of 4 patients, in which CD45⁻ cell sorting was performed to enrich structural cell populations. After sequencing, more than 10,000 single cells were obtained, covering epithelial cells, fibroblasts, endothelial cells, and myofibroblasts. This dataset was thus suitable for evaluating the expression characteristics of the target genes in non-immune stromal and structural cells.

GSE145281 included peripheral blood mononuclear cell (PBMC) samples from 7 bladder cancer patients undergoing immunotherapy. This dataset profiled over 20,000 immune cells, including CD4⁺ and CD8⁺ T cells, B cells, NK cells, and monocyte/macrophages, which enabled the assessment of gene expression in immune cell compartments.

The processed expression matrices and cell-type annotation files were obtained from the TISCH2 platform, and data analysis was performed using Seurat (v4.3.0) and MAESTRO in R (v4.3.2). After quality control, normalization, and clustering, t-SNE dimensionality reduction was performed, and FeaturePlot and VlnPlot functions were used to visualize expression trends across cell types. The AverageExpression function was further applied to calculate the mean expression level of each key gene across major cellular subpopulations.

## Validation of protein expression levels

To further validate the expression of key genes at the protein level, this study systematically assessed the protein expression patterns of ENO1, GTF2F1, SPR, and WBP11 in bladder cancer tissues with the help of the Human Protein Atlas (HPA, https://www.proteinatlas.org/) public database. The HPA database provides spatial expression information of proteins in human tissues through standardized immunohistochemical staining experiments with highly reproducible and authoritative histological images taken by automated microscopy and reviewed manually. We used gene names to retrieve target proteins in the HPA database separately, corresponding to localization and expression intensity analysis in cancerous specimens of bladder tissue. All image sources were bladder cancer tissue sections from patients with uroepithelial carcinoma, and the most representative images of expression were selected for each gene for presentation. Staining results were graded and assessed by antibody staining intensity (strong, weak, not detected), staining location (cytoplasmic, membranous, nuclear), and percentage of positive cells (<25%, 25–75%, > 75%). In addition, to ensure the objectivity of the interpretation, we combined the staining intensity and distribution range to make a comprehensive judgment of the expression level. To further enhance the clarity and presentation of the images, all images were locally enlarged based on the original resolution (the following figure shows the locally enlarged image), to observe the cytoarchitectural features and localization differences of the stained regions, and finally the expression distribution characteristics of each gene were demonstrated in the figure by pairing a high-magnification view with a low-magnification panoramic view. This part of the analysis helps to bridge the potential differences between the transcriptome level and the protein level, providing reliable experimental evidence for the functional study of key genes and the value of biomarkers.

## Molecular Docking

In this study, the binding ability between target proteins and small-molecule compounds was evaluated by a systematic molecular docking approach. First, the 2D structural data of the target ligands were downloaded from the PubChem database (https://pubchem.ncbi.nlm.nih.gov) [35] and converted into 3D structural models with the help of ChemOffice software, and finally saved as mol2 format files. Next, protein crystal structures corresponding to the target genes with high resolution were screened as receptors from the RCSB PDB database (https://www.rcsb.org/) [36], and the crystal structures were pre-processed, including the removal of water molecules, cofactors, irrelevant ions, and other heteroatoms, with the help of PyMOL software [37], and saved as PDB format. Subsequently, docking simulations were performed using AutoDock Vina 1.5.6 software [38]. In Autodock Tools, the protein and small molecule structures were hydrogenated and assigned Gasteiger charges, the original water molecule structure was removed, and rotational bonding degrees of freedom were set for the ligands, respectively. For the docking parameters, the binding site region was defined, a grid box was constructed, and the center coordinates and grid size were set to ensure that the potentially active pockets of the protein were covered. After docking, the conformation with the lowest binding affinity was selected as the optimal model based on the Vina score (binding affinity). To further elucidate the specific binding mode between the small molecule and the target, three-dimensional structures were displayed with the help of PyMOL and Discovery Studio Visualizer 2019 software, and two-dimensional force diagrams were drawn to visualize the important mechanical relationships, such as hydrogen bonding, hydrophobic interactions, and aromatic ring stacking. In general, a binding energy lower than −5.0 kcal/mol indicates that the ligand and the receptor have good binding activity; if it is lower than −7.0 kcal/mol, it indicates that the two have strong affinity and stable binding conformation. The lower the binding energy, the more stable and biologically active the ligand-protein binding is.

## Molecular dynamics simulation

To further evaluate the stability and interaction mechanism of Bisphenol A (BPA) in complex with key target proteins, 100 ns molecular dynamics simulations were performed in this study using the GROMACS 2022 software package [39]. The protein topological parameters used are derived from the CHARMM36 force field to ensure accurate modeling of

interatomic interactions in complex biomolecular systems, while the topology of the small-molecule ligand, BPA, is constructed based on the General AMBER Force Field 2 (GAFF2), which takes into account its flexible structural properties and electron distribution. In the initial preparation stage, the protein-ligand complex was placed in an adequately sized three-dimensional cubic simulation box, and the minimum distance between the box boundary and the outermost atoms of the protein was set to 1.2 nm to avoid interference between periodic images. The system was solvated using the TIP3P water model to construct an aqueous system that more closely resembles the physiological environment. Subsequently, an appropriate amount of $Na^+$/Cl- ions was added to neutralize the overall charge of the system to form an electrically neutral system. The energy minimization step was performed using the Steepest Descent algorithm to ensure that the system was in a reasonable energy conformation before entering the kinetic simulation. Next, the system was pre-equilibrated in an isothermal isovolumic system (NVT) and an isothermal isobaric system (NPT) for 100 ps each, and the temperature and pressure of the system were maintained at 310 K and 1 bar using a V-rescale temperature coupler and a Parrinello-Rahman pressure coupler, respectively. Electrostatic interaction calculations were performed using the Particle Mesh Ewald (PME) method, and the truncation radius of van der Waals force and Coulomb force is uniformly set to 1.0 nm to balance the computational efficiency and accuracy. The neighborhood search algorithm adopts the Verlet list updating scheme. Finally, 100 ns molecular dynamics simulations of the production period were performed based on the system after equilibrium was completed. The trajectory data were used to further calculate the root mean square displacement (RMSD), radius moment (Rg), solvent accessible surface area (SASA), number of hydrogen bonds (H-bonds), and residue fluctuation (RMSF), and to analyze the distribution of the stable conformations of the complexes by Free Energy Landscape, to comprehensively reveal the kinetic behaviors and structural features of the binding of BPA to the target protein. The kinetic behavior and structural characteristics of BPA binding to target proteins were comprehensively revealed.

## Immunofluorescence staining

Five paired bladder cancer and adjacent normal tissue samples were collected from patients who underwent surgical resection at the Department of Urology, Yantai Yuhuangding Hospital. Paraffin-embedded sections were deparaffinized, rehydrated, and subjected to antigen retrieval in citrate buffer (pH 6.0). After blocking with 5% bovine serum albumin (BSA) for 30 min at room temperature, the sections were incubated overnight at 4 °C with a primary antibody against ENO1 (1:200, Proteintech, Wuhan, China). The slides were then washed and incubated with an Alexa Fluor 488–conjugated secondary antibody (1:500, Proteintech, Wuhan, China) for 1 h at room temperature. Nuclei were counterstained with DAPI (Beyotime, Shanghai, China) for 5 min. Images were captured using a fluorescence microscope (Leica Microsystems, Germany), and fluorescence intensity was quantified by ImageJ software.

## Expression validation

Three paired bladder cancer tissues and adjacent normal tissues from the same cohort were used for quantitative real-time PCR (qPCR) validation. Total RNA was extracted from tissue samples using a standard RNA extraction protocol, and complementary DNA (cDNA) was synthesized by reverse transcription according to the manufacturer's instructions. qPCR was performed using gene-specific primers for WBP11, SPR, and GTF2F1, with GAPDH as the internal control. Relative mRNA expression levels were calculated using the $2^{-\Delta\Delta Ct}$ method. The primer sequences were as follows: WBP11 forward, CCTTCTCAGATACAAGCACCTCC; reverse, AGGTGGTCTCAGGAATGGAGGA. SPR forward, TGCAGGAAAGGCTGCTCGTGAT; reverse, TGCTGCATGTCTGTGTCCAGAG. GTF2F1 forward, GAGGTGGACTA CATGTCAGACG; reverse, ACTCTCCTCACTACTGTCGCTC.

Three paired bladder cancer and adjacent normal tissues from the same cohort were used for Western blot validation. Total protein was extracted using RIPA buffer containing protease inhibitors. Equal amounts of protein (30 µg per lane) were separated on 10% SDS–PAGE and transferred to PVDF membranes. After blocking with 5% non-fat milk for 1 h, membranes were incubated overnight at 4 °C with anti-ENO1 (1:1000, Proteintech) and anti-GAPDH (1:5000, Proteintech)

antibodies. After washing, membranes were incubated with HRP-conjugated secondary antibodies (1:5000, Proteintech) for 1 h at room temperature. Protein bands were visualized by enhanced chemiluminescence (ECL, Thermo Fisher), and relative densitometric analysis was performed using ImageJ.

## Results

### WGCNA analysis and intersecting gene screening

The flowchart of this study is shown in Fig 1. To identify co-expression modules associated with bladder cancer and prioritize BPA- and lactylation-related candidate genes, we first performed WGCNA using the GSE13507 dataset. Based on the normalized expression matrix of the GSE13507 dataset, 7,735 highly variable genes ranked within the top 30% of variability were selected for WGCNA. Sample clustering results showed that no obvious abnormal samples were found (Fig 2A), and all of them were included in the subsequent network construction. To construct a scale-free network, we evaluated the topological fit under different soft-thresholding powers and selected a power of 14, at which the network most closely approximated a scale-free distribution (R²=0.88; Fig 2B). The similarity between genes was calculated based on the Topological Overlap Matrix (TOM) and the initial modules were identified by the Dynamic Tree Cut algorithm. The clustering results identified a total of 17 co-expression modules (Fig 2C), excluding unclassified genes (gray modules). Pearson correlation analysis between each module and clinical phenotype (bladder cancer vs. normal) showed that several modules were significantly associated with disease status. Among them, the blue, yellow, and green modules showed the strongest correlations (|cor|>0.3, P<0.05; Fig 2D), suggesting their potential relevance to bladder cancer progression. A total of 3651 module genes were extracted from the above significantly related modules as a candidate set for subsequent functional analysis.

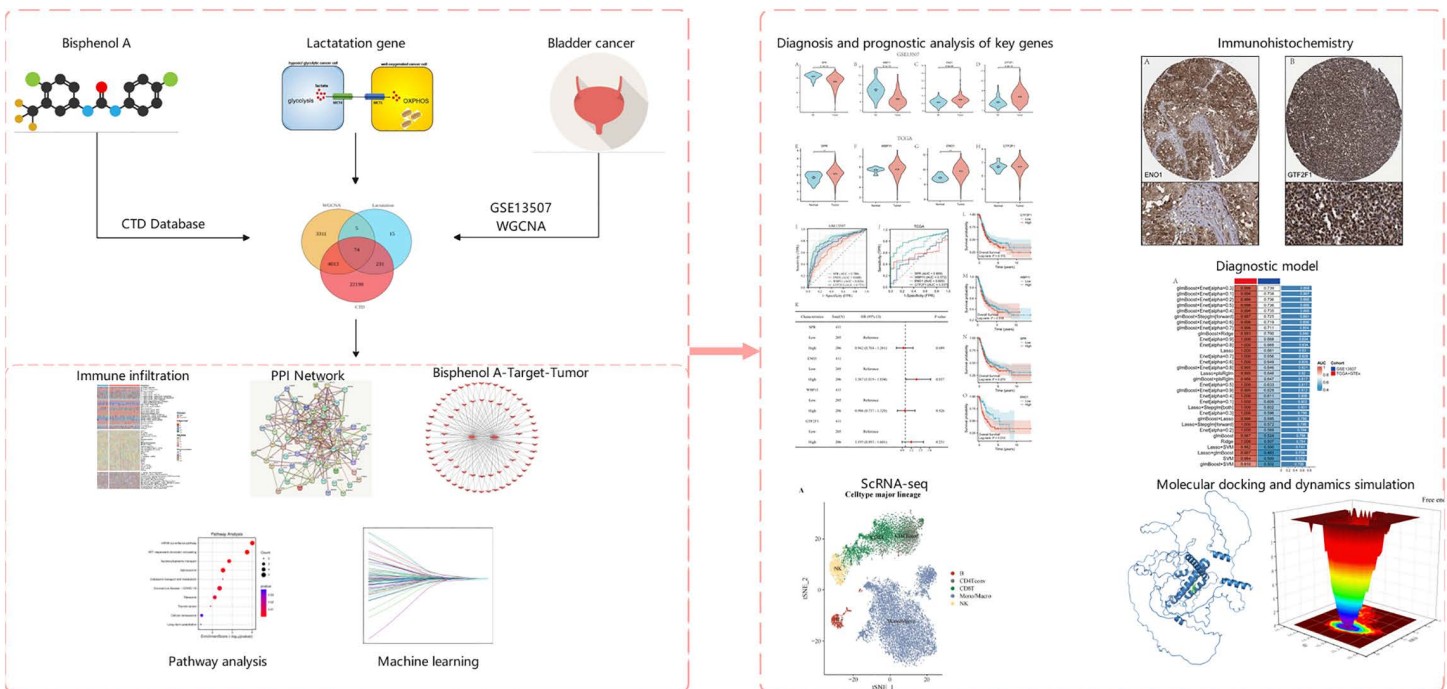

**Fig 1. Flowchart of this study.** Bisphenol A is presented as a possible cause of bladder cancer by acting on lactylation-related genes.

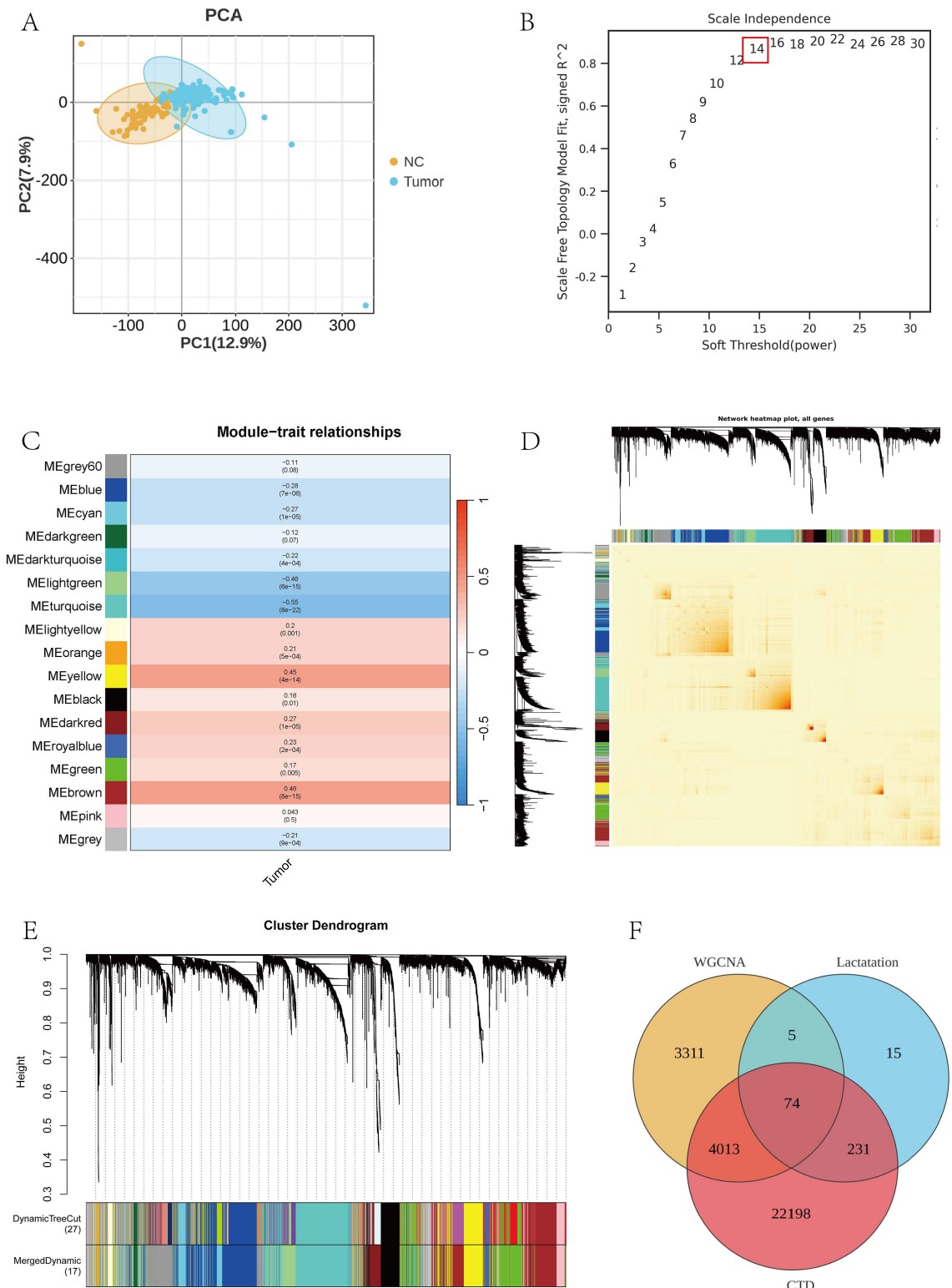

**Fig 2. WGCNA co-expression module construction and intersection gene screening. (A)** Sample clustering tree diagram for detecting abnormal samples; **(B)** Scale-free topological fit and average connectivity analysis under different soft thresholds, selecting power = 14; **(C)** TOM-based module identification clustering tree, with the colors representing different modules; **(D)** Heatmap of the correlation between modules and clinical phenotypes

(bladder cancer vs. normal); **(E)** Heatmap of correlations between BPA targets, lactylation genes and WGCNA Venn diagram of module genes, intersected to obtain 74 candidate key genes.

To further refine the candidate set, BPA-related target genes were retrieved from the CTD database, yielding 26,516 genes. In addition, 328 lactylation-related genes were collected from the published literature, including lactylated substrate proteins, regulatory enzymes, and functionally related factors. Intersection analysis of these three gene sets identified 74 overlapping genes with co-expression, BPA-targeting, and lactylation-related features (Fig 2E–F), providing the basis for subsequent candidate gene prioritization and downstream analyses. The observed 74-gene intersection was significantly larger than expected under random sampling conditions.

## Analysis of immune infiltration

Quantitative analysis of major immune cell types using the MCPcounter algorithm revealed (Fig 3A) that estimated cytotoxic lymphocyte infiltration levels were higher in the primary tumor group compared with normal and peritumoral tissues, while the recurrence group showed a modest reduction relative to primary tumors but remained elevated compared with normal tissue. CD8$^+$T cells exhibited a similar relative trend across groups.

CIBERSORT-based immune deconvolution analysis (Fig 3B) indicated that estimated M2 macrophage fractions were higher in recurrent tumors than in primary tumors and peritumoral tissues, while regulatory T cells showed increased relative abundance in tumor samples compared with normal tissue. These patterns were consistent with an immunosuppressive tumor microenvironment.

ssGSEA analysis of immune-related gene sets (Fig 3C) showed differential enrichment of NK cell–related signatures across tissue groups, with higher enrichment in peritumoral samples and lower enrichment in primary tumor tissues. These pathway-level scores represent relative immune activity patterns rather than direct measurements of immune cell function.

Consistency analysis across the three algorithms showed concordant enrichment trends for cytotoxic lymphocytes, CD8$^+$T cells, and M2 macrophages across different methods. This cross-method concordance supports the robustness of the inferred immune infiltration patterns.

Correlation analyses within each algorithm (Fig 4A–C) further revealed coordinated patterns among immune cell subsets, including positive correlations among adaptive immune cell populations and inverse relationships between macrophage subtypes. In addition, the four prioritized candidate genes (SPR, WBP11, ENO1, and GTF2F1) showed consistent associations with multiple immune cell signatures across all three algorithms. These associations indicate potential links between candidate genes and immune infiltration patterns, but do not establish direct regulatory or causal relationships.

Overall, the multi-algorithm immune infiltration analysis revealed reproducible and consistent immune-related trends in bladder cancer tissues. These findings should be interpreted as computationally inferred immune landscape patterns that generate hypotheses regarding tumor–immune interactions within a BPA-informed toxicogenomics framework, rather than as precise measurements or mechanistic evidence.

## Expression characterization and functional network analysis of intersecting genes

To characterize the expression patterns of the intersecting genes in bladder cancer, their expression profiles were extracted from the GSE13507 dataset and visualized using a heatmap to compare normal and tumor samples. Most intersecting genes showed a trend toward higher expression in tumor tissues, and several genes, including EAF1, UPF1, KHDRBS1, and SP3L, were markedly upregulated in the tumor group (Fig 5A), indicating preferential upregulation of these genes in bladder cancer tissues. To explore potential interactions among the intersecting genes, a protein–protein interaction (PPI) network was constructed using the STRING database. After removal of unconnected nodes, the network contained 74 nodes and 216 edges, with an average node degree of 5.84 and an average clustering coefficient of 0.487.

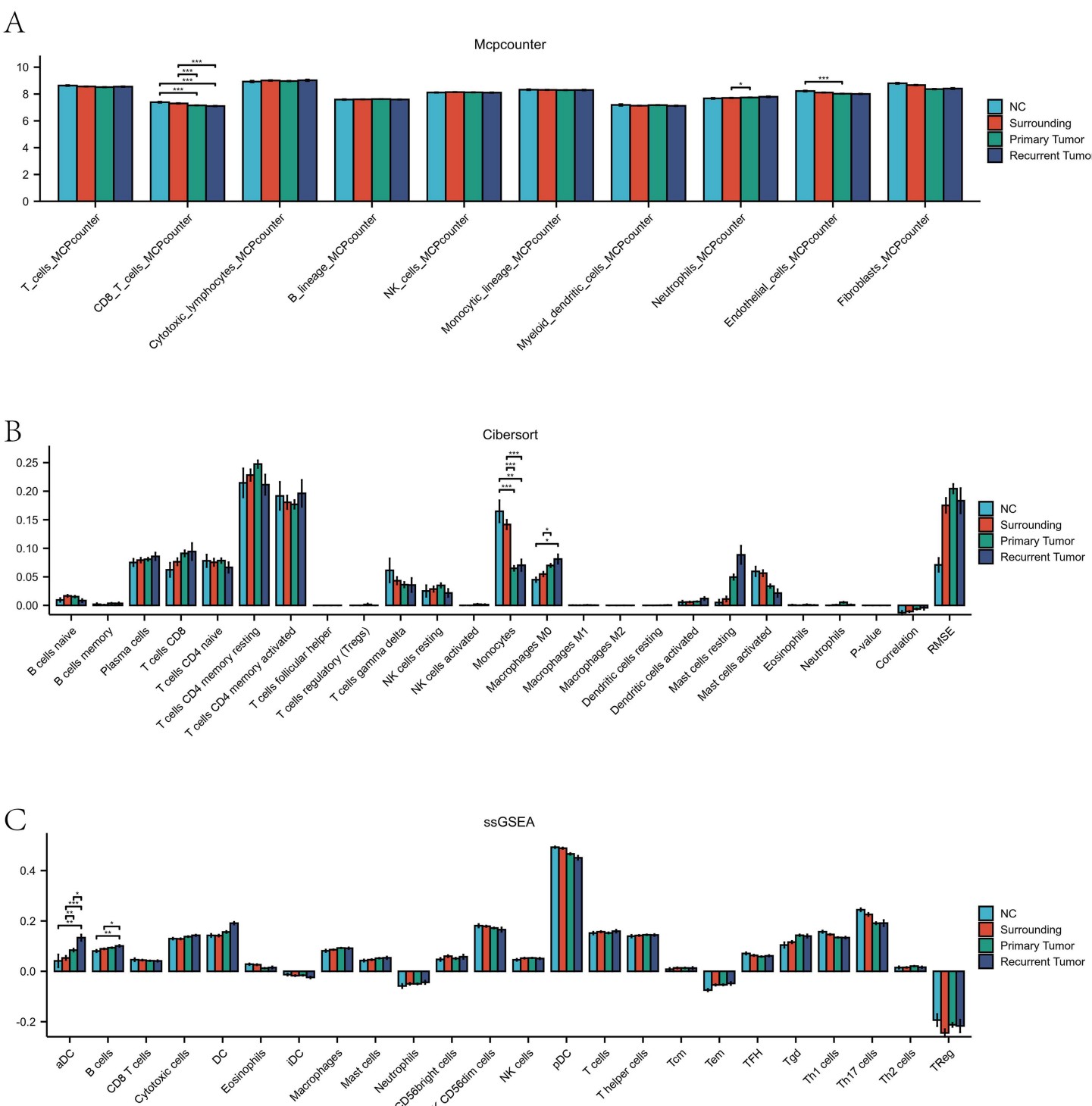

**Fig 3. Immune infiltration profiles across subgroups using multiple algorithms(A) MCPcounter analysis: Grouped bar plots depicting absolute infiltration scores of major immune cell types in normal bladder mucosa (NC), paracancerous tissues (Surrounding), primary bladder tumor, and recurrent bladder tumor groups.** Error bars indicate standard deviation. **(B)** CIBERSORT analysis: Peak – shaped histograms showing relative proportions of 22 immune cell subsets across the four groups, highlighting immune subpopulation heterogeneity. (C) ssGSEA analysis: Grouped bar plots presenting enrichment scores of immune cell – associated gene sets, reflecting immune cell activity changes among groups.

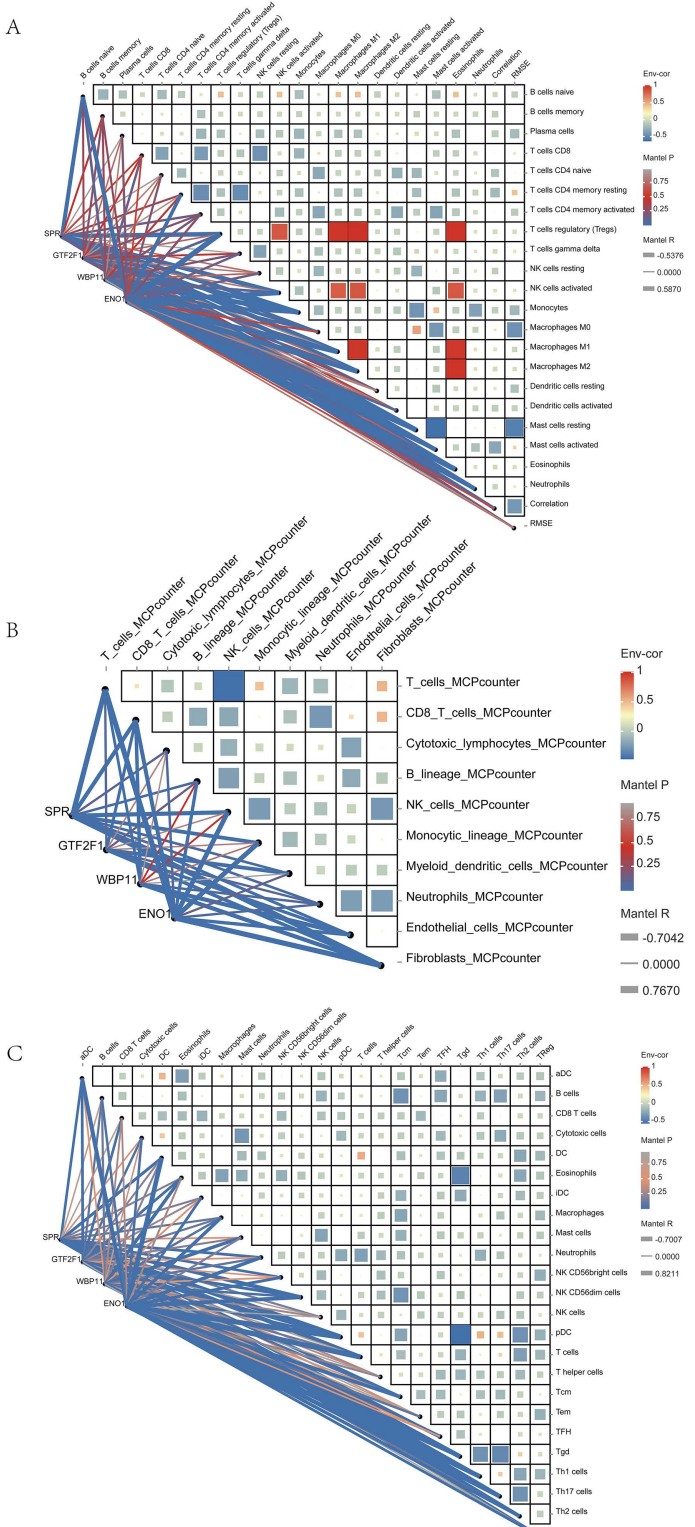

**Fig 4. Correlation analysis between immune cell infiltrations.** Four key genes showed strong correlation with multiple immune cells in three different algorithms for immune infiltration analysis. Line thickness indicates the strength of correlation, line color indicates the degree of significance, color block color within each circle in the heatmap indicates positive or negative correlation coefficient, and color block size indicates the size of absolute value of correlation coefficient.

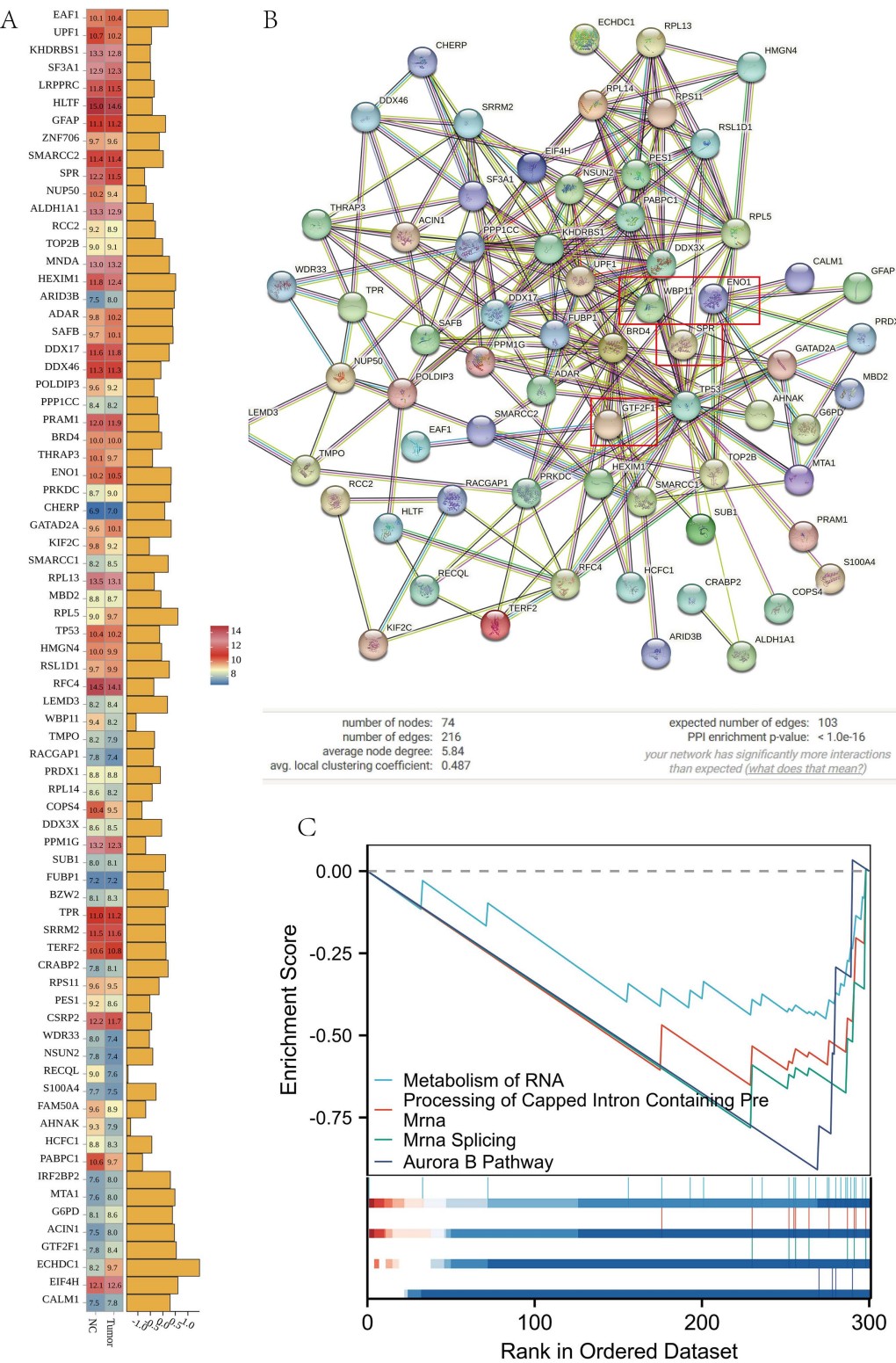

Fig 5. Expression characterization and functional network analysis of intersecting genes. (A) Heatmap displaying the expression patterns of the 74 intersecting genes across samples in the GSE13507 dataset. Each row represents a gene, and each column represents a sample. The color scale from blue to red indicates low to high Z-score normalized expression levels, Bar chart: The abscissa represents the corresponding difference fold size,

and the higher the bar, the greater the difference fold; **(B)** Protein-protein interactions (PPI) network graph constructed by the STRING database, with nodes indicating the gene products and the connecting lines indicating the functional associations; and **(C)** Enrichment score (ES) curves for pathways including "Metabolism of RNA", "Processing of Capped Intron Containing Pre - mRNA", "mRNA Splicing", and "Aurora B Pathway" during GSEA. The lower panel shows the distribution of genes in these pathways within the phenotype – ranked gene list.

PPI enrichment analysis was highly significant (P < 1.0e-16; Fig 5B), indicating that the observed interaction density was greater than expected by chance and suggesting coordinated involvement of these genes in bladder cancer–related pathways. To investigate the biological processes and signaling pathways associated with these genes, we further performed GSEA. GSEA revealed significant enrichment of several pathways, including "Spliceosome" (NES = 2.12, FDR < 0.001), "RNA degradation" (NES = 1.98, FDR < 0.001), and "mRNA surveillance pathway" (NES = 1.85, FDR < 0.005), suggesting that these intersecting genes were mainly enriched in post-transcriptional regulatory pathways (Fig 5C). In this study, we constructed a BPA-target-tumor network topology, which contained 76 nodes with a network density of 0.052, indicating sparse connections between nodes(Fig 6). The network heterogeneity is 2.959, suggesting that there are a few key nodes with high connectivity in this network, which may play a central regulatory role in the network. The network is structurally complete without isolated nodes, self-connections, and multiple edges. The analysis time is 0.029 seconds.

## Machine learning screening of key genes

To identify core feature genes with potential diagnostic value for bladder cancer, three machine learning methods— LASSO, SVM-RFE, and random forest—were applied to the intersecting genes for feature selection. Ten-fold cross-validation of the LASSO model identified the minimum mean squared error at lambda.min = 0.01 (Fig 7A). At this value, 12 feature genes with non-zero regression coefficients were retained (Fig 7B). The SVM-RFE analysis had the smallest cross-validation error (0.133) and the highest accuracy (0.867) at a feature number of 10, corresponding to the selection of 10 optimal feature genes (Fig 7C-D). The random forest model calculated variable importance based on the Gini index, and extracted the top 30 genes with the average reduction (Fig 7E), which included several key regulators such as TOP2B, WBP1, RPL14, UPF1, AHNAK, and so on. Finally, integration of the three feature-selection results using a Venn diagram identified four overlapping genes—ENO1, WBP11, GTF2F1, and SPR—as the final candidate genes (Fig 7F). These genes were prioritized for further analysis as BPA- and lactylation-associated candidates in bladder cancer.

## Construction and validation of diagnostic models

We systematically compared the diagnostic performance of 28 candidate models in the training set (TCGA+GTEx) and validation set (GSE13507). As shown in Fig 8A, most models achieved excellent classification ability in the training cohort, with AUC values close to 1.0. When validated in the independent GSE13507 dataset, the performance decreased but remained robust, with several models retaining satisfactory discrimination. Among them, the glmBoost+Enet[α = 0.3] model demonstrated the best balance, with an AUC of 0.996 in TCGA+GTEx and 0.739 in GSE13507, while the external validation AUC of individual models generally ranged from 0.70 to 0.87. These findings indicate that although the models tend to overfit in the training set, they still possess cross-cohort generalization ability.

The representative ROC curves for the glmBoost+Enet[α = 0.3] model are shown in Fig 8B, with an AUC of 0.996 in TCGA+GTEx and 0.739 in GSE13507, confirming its diagnostic reliability. This phenomenon is likely attributable to the high dimensionality of transcriptomic features relative to the sample size and the strong correlation structure among genes, which can lead to overly optimistic performance in the training cohort. To minimize information leakage, all feature selection procedures and model optimization steps were performed exclusively within the training set, while the validation cohort was kept completely independent and was not involved in any stage of feature selection or parameter tuning.

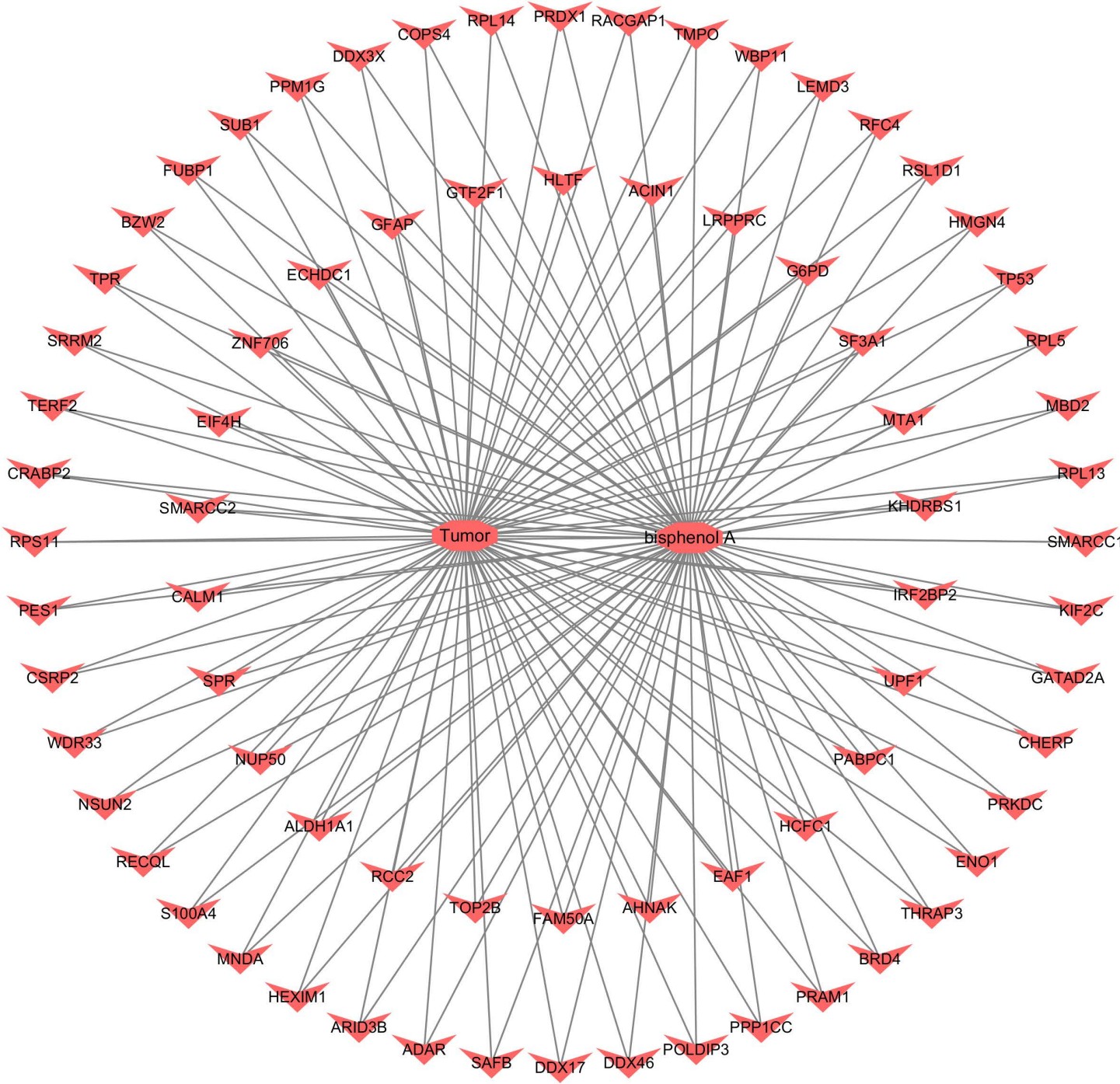

**Fig 6. The "BPA-target-tumor" network.** The network consists of 76 nodes with a density of 0.052 and a heterogeneity index of 2.959. No isolated nodes, self-loops, or multi-edge pairs were observed. The network is fully connected (1 connected component), with a diameter and radius of 2, indicating a compact structure. A high network centralization value (0.96) and characteristic path length (1.948) suggest the presence of key hub nodes. The clustering coefficient is 0.0, and the average number of neighbors per node is 3.895. A total of 5700 shortest paths were calculated, covering 100% of all node pairs.

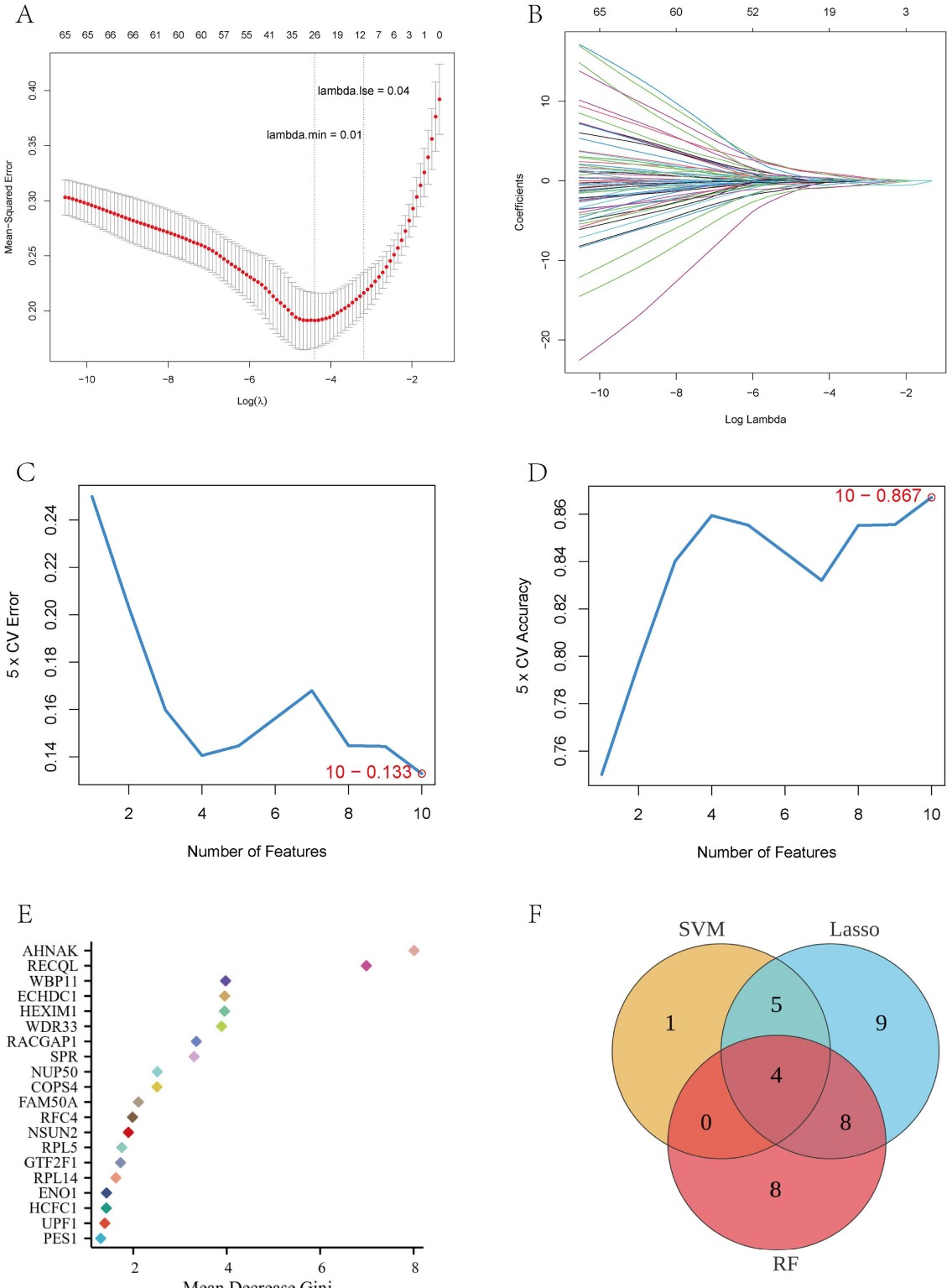

**Fig 7. Machine learning algorithm to screen key genes. (A, B)** LASSO regression cross-validation to determine the optimal lambda value and screen for non-zero coefficient genes; **(C, D)** SVM-RFE algorithm evaluating cross-validation error vs. accuracy with different number of features to determine the optimal set of genes; **(E)** Mean-reduction Gini coefficient-based assessment of variable significance in the Random Forest model; **(F)** Venn diagrams of the intersection of three algorithms for screening results. identifying four common candidate key genes.

A

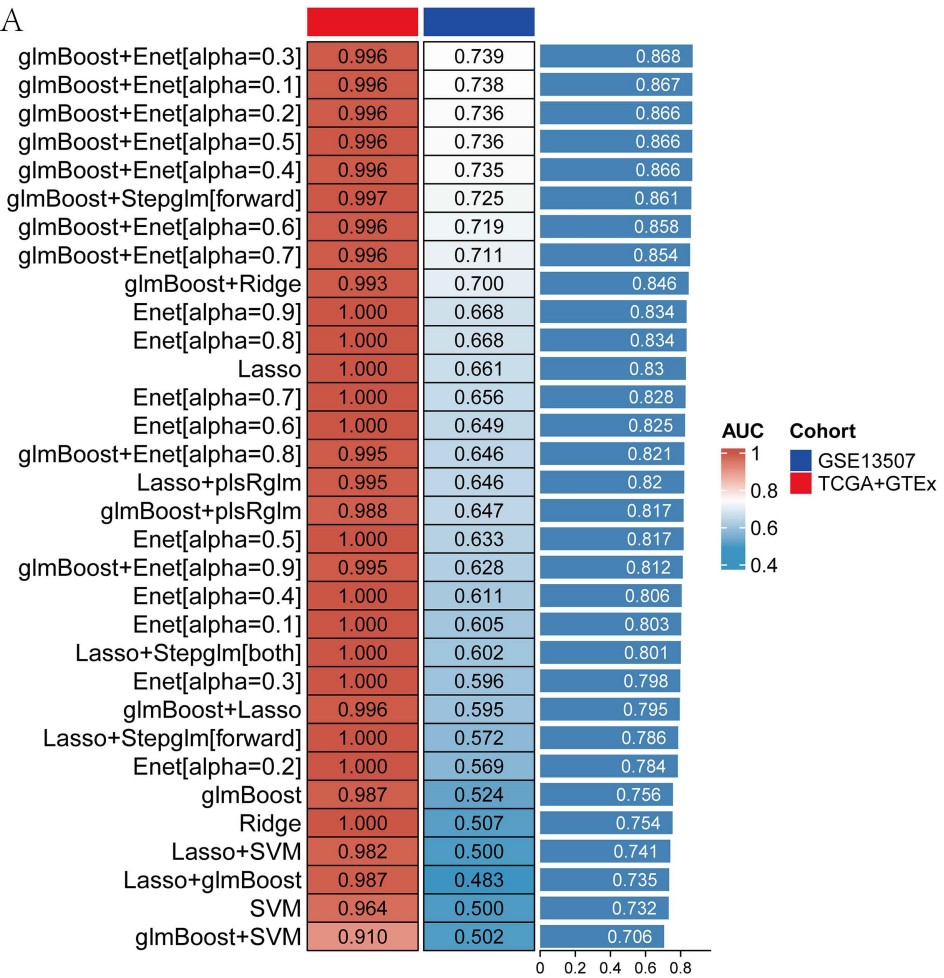

B

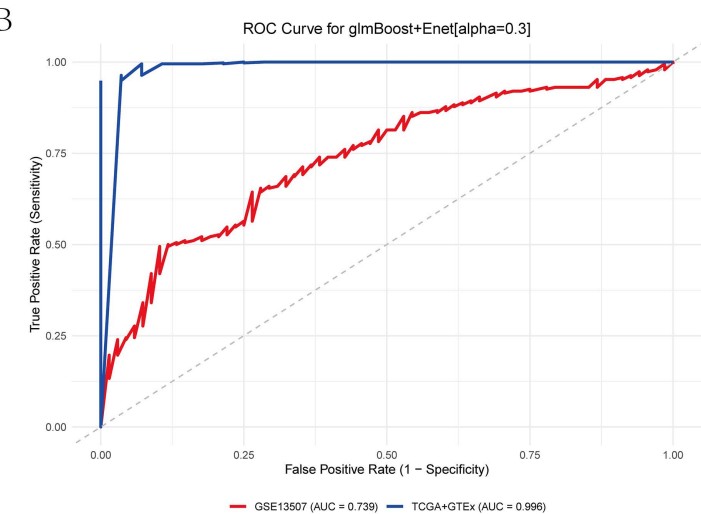

**Fig 8. Diagnostic performance evaluation of multiple machine learning models in the training and validation sets. (A)** AUC values of 28 machine learning models constructed in the TCGA+GTEx training set (red) and validated in the GSE13507 dataset (blue). Models are sorted by validation AUC, showing that several algorithms maintain robust performance across datasets.(B) ROC curves of the glmBoost+Enet[α=0.3] model in the TCGA+GTEx training set (blue, AUC=0.996) and the GSE13507 validation set (red, AUC=0.739).

## Expression, diagnostic efficacy and prognostic analysis of key genes

In the GSE13507 cohort, the violin plot results showed that the expression levels of the four key genes, SPR, WBP11, ENO1 and GTF2F1, were significantly higher in bladder cancer tissues than in normal tissues (all P less than 0.001, Fig 9A-D). However, in the TCGA-BLCA dataset, WBP11 and ENO1 retained the upregulation trend observed in GSE13507 (Fig 9F,G), whereas SPR and GTF2F1 showed the opposite pattern, with higher expression in normal tissues (Fig 9E,H). These findings suggest that the expression patterns of some genes varied across datasets, potentially owing to differences in sample sources, platform technologies, or grouping criteria. ROC analysis showed that in GSE13507, ENO1 had the highest AUC (0.868), and the AUCs of WBP11, SPR and GTF2F1 were 0.820, 0.788 and 0.755, respectively, which all showed good diagnostic efficacy (Fig 9I). In the TCGA validation set, ENO1 still showed strong discriminatory ability (AUC = 0.803), while the AUCs of SPR, WBP11, and GTF2F1 were 0.699, 0.572, and 0.537, respectively (Fig 9J), suggesting that the discriminatory ability of this gene combination across datasets was somewhat stable, with ENO1 showing the most prominent performance in particular. Further survival analysis showed that only high ENO1 expression was significantly associated with poorer overall survival in the TCGA-BLCA cohort (HR = 1.367, 95% CI: 1.019–1.834, P = 0.037), whereas the other three genes did not reach statistical significance in Cox regression (Fig 9K). Kaplan–Meier analysis further showed that the ENO1 high-expression group had significantly poorer overall survival than the low-expression group (log-rank P = 0.010), whereas no significant survival differences were observed for SPR, WBP11, or GTF2F1 (Fig 9L–O). To further support the relevance of WBP11, SPR, and GTF2F1 in bladder cancer, additional survival analyses were conducted across multiple independent cohorts (GSE154261, GSE69795, GSE31684, GSE19423, GSE39281, IMvigor210, and GSE13507), showing that these genes also displayed prognostic associations in supplementary datasets (S1 Fig). These results suggest that ENO1 may play a critical role in the diagnosis and prognosis of bladder cancer.

## Localization analysis of key genes

To further clarify the potential functions of key lactylation-related genes in the microenvironment of bladder cancer tissues, we performed expression localization analyses in two single-cell transcriptome datasets, GSE130001 and GSE145281. Fig 10 shows the expression distribution of the four key genes (ENO1, WBP11, GTF2F1, and SPR) in the GSE130001 dataset together with cell-type annotations. The dataset mainly included epithelial cells, endothelial cells, fibroblasts and myofibroblasts. The results showed that ENO1 was mainly expressed in epithelial cells; WBP11 was widely distributed in all cell types, especially in epithelial cells; GTF2F1 was weakly expressed with obvious limitation; SPR was mainly expressed in endothelial cells and fibroblasts at low to medium levels. Fig 11 further demonstrates the localization of GSE145281 in the immune microenvironment dataset, which covers immune cells such as T cell subsets, monocytes/macrophages, NK cells and B cells. The results showed that ENO1 was actively expressed in CD8$^+$T cells and monocyte/macrophages, WBP11 was relatively highly expressed in B cells and macrophages, and GTF2F1 and SPR were mainly expressed in NK cells and monocytes, with an overall low expression level. Combined analysis of the average expression profiles from the two datasets (Fig 12) showed that ENO1 expression in structural cells was concentrated in epithelial cells, whereas in immune cells it was mainly distributed in CD8$^+$T cells and monocyte/macrophage populations. WBP11 showed a broader distribution across both datasets, suggesting potential involvement in multiple cellular contexts. By contrast, GTF2F1 and SPR were generally expressed at low levels and showed a degree of cell-subpopulation specificity.

## Validation of protein expression levels

Analysis of immunohistochemical images from the HPA database showed that the four key genes were expressed at different protein levels in bladder cancer tissues. ENO1 (Fig 13A) showed strong positive staining in bladder cancer tissues, mainly localized to the cytoplasm and parts of the membrane, with high staining intensity and more than 75% positive cells. GTF2F1 (Fig 13B) and WBP11 (Fig 13D) both showed cytoplasmic staining with partial nuclear localization, with moderate-to-strong staining intensity and 25%–75% positive cells. Their staining distribution appeared relatively focal and

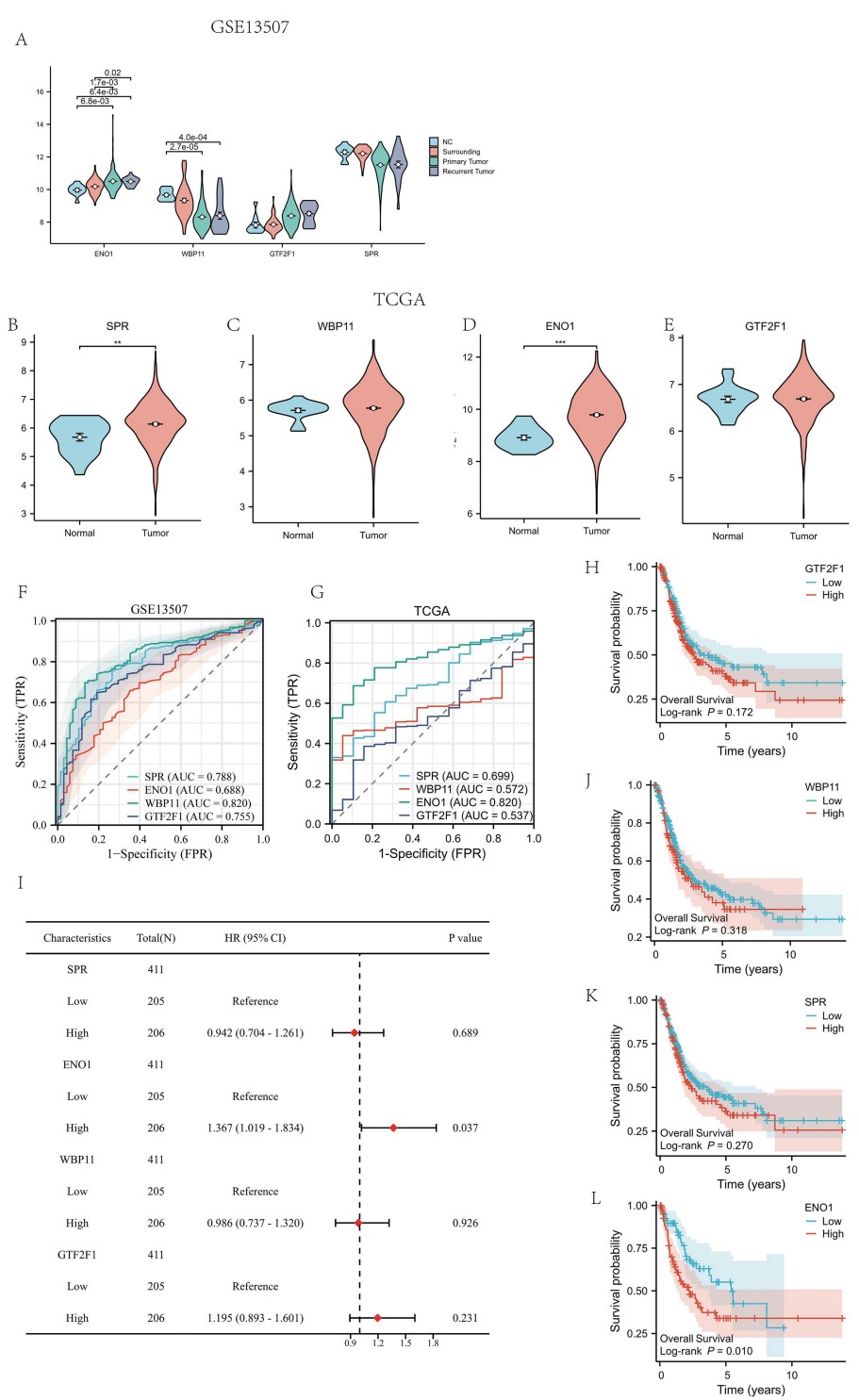

**Fig 9. Expression, diagnostic efficacy and prognostic analysis of four key genes in different cohorts. (A-E)** Violin plots showing differential expression of key genes between tumor and normal samples in TCGA and GEO cohorts. Statistical significance was assessed using Welch's t-test, and P-values were adjusted for multiple testing using the Benjamini–Hochberg (BH) correction. Adjusted P-values are indicated in the figure (adjusted P < 0.05 considered significant); **(F,G)** ROC curves and AUC values of the four genes in the two datasets; (I) one-way Cox regression analysis of the four genes in the TCGA cohort; (H,J-L) Kaplan-Meier survival curves showed that only the ENO1 high-expression group had a significantly worse overall survival (P = 0.010).

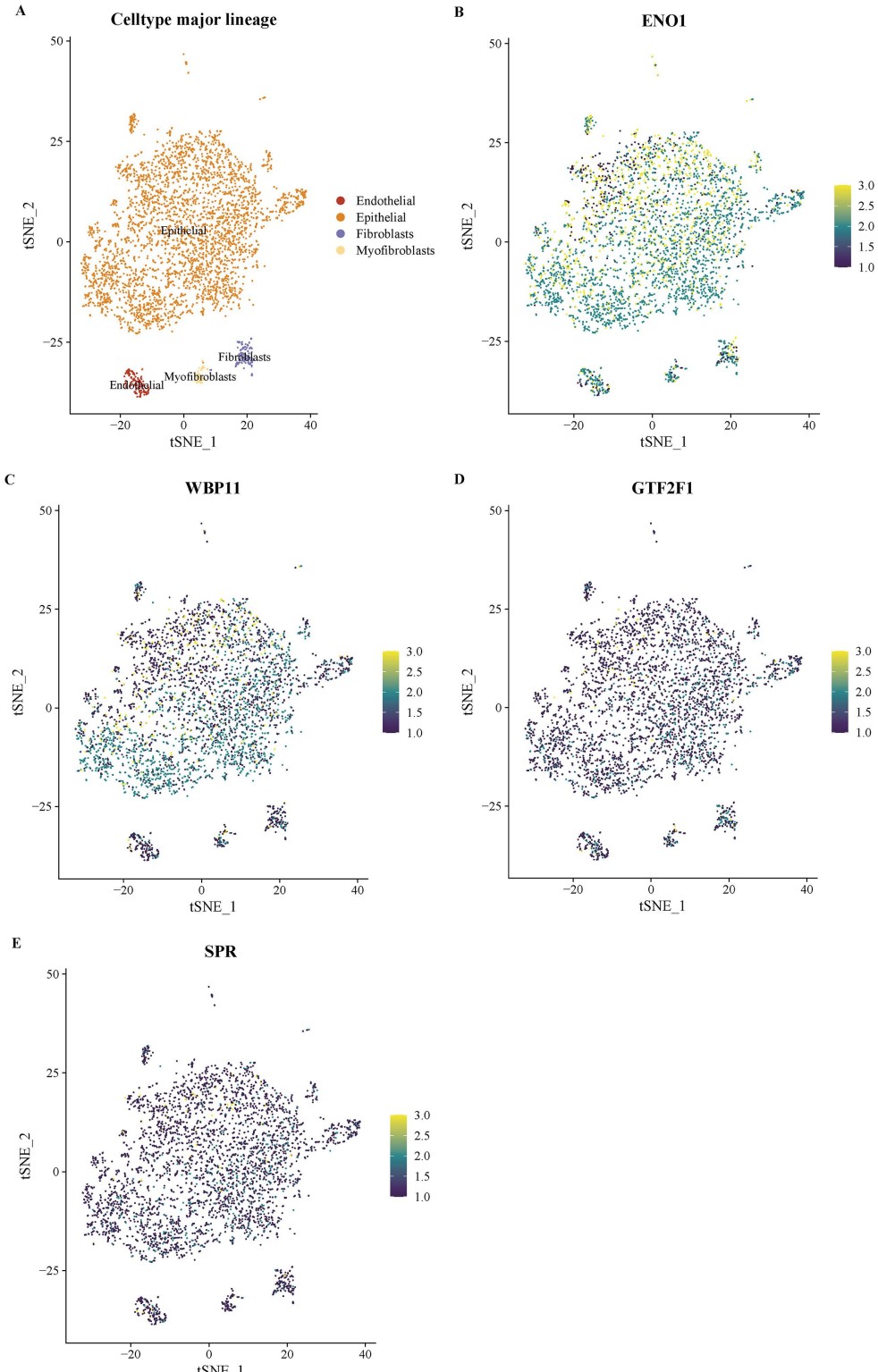

**Fig 10. Characterization of cellular subpopulation localization of key genes in the GSE130001 dataset.** (A) t-SNE plot showing major cell types including epithelial cells, fibroblasts, endothelial cells and myofibroblasts. **(B-E)** Showing the expression distribution of ENO1, WBP11, GTF2F1 and SPR in different cell types, respectively, with darker colors indicating higher expression levels.

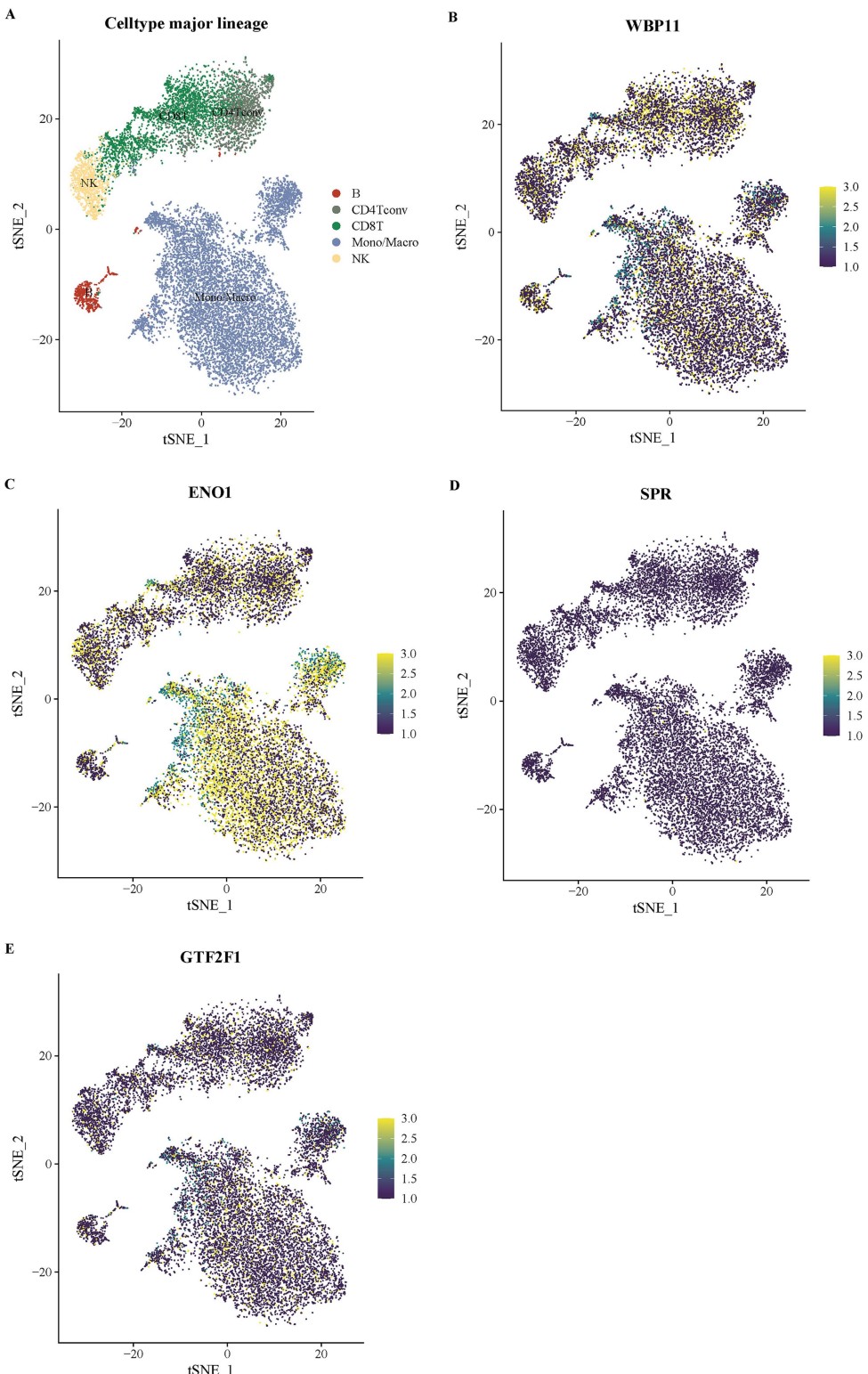

**Fig 11. Distribution of expression of key genes in immune cell subpopulations in the GSE145281 dataset.** (A) t-SNE plot demonstrating clustering of immune-related cell types including CD4+T cells, CD8+T cells, NK cells, B cells, and monocyte macrophages. **(B-E)** Expression plots of WBP11, ENO1, SPR, and GTF2F1 in immune cell populations, respectively, with expression intensity indicated by color gradient.

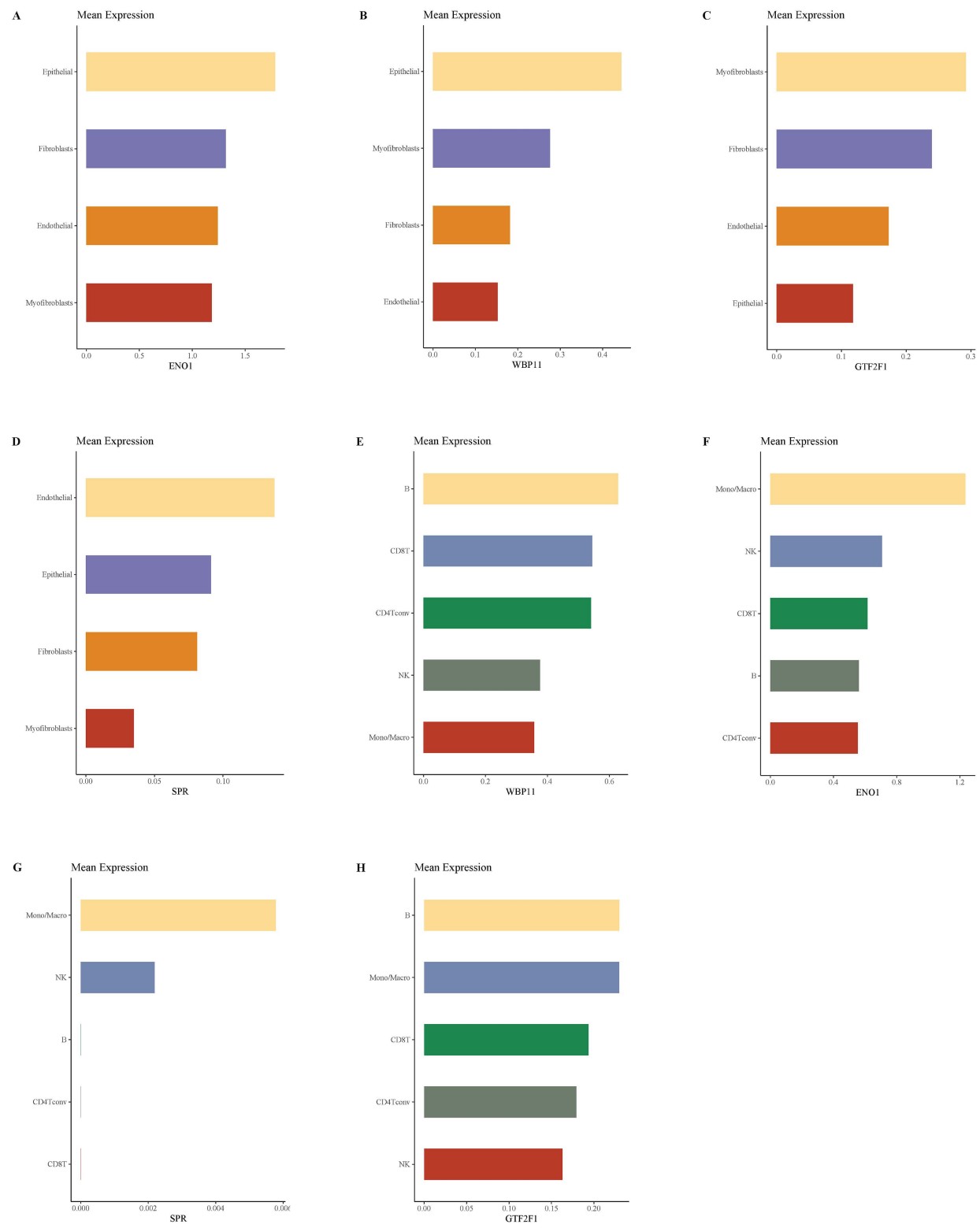

**Fig 12. Histogram of average expression of four key genes in different cell subpopulations. (A-D)** Shows the average expression levels of ENO1, WBP11, GTF2F1 and SPR in the GSE130001 structural cell population. **(E-H)** Shows the average expression of the same gene in the GSE145281 immune cell population.

A

ENO1

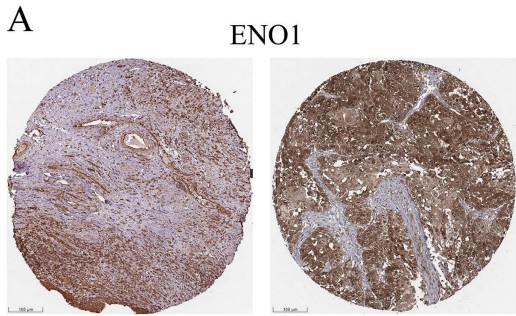

B

GTF2F1

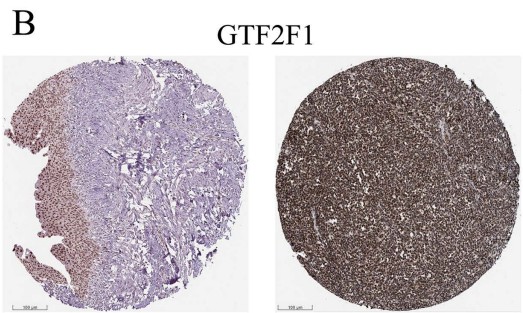

C

SPR

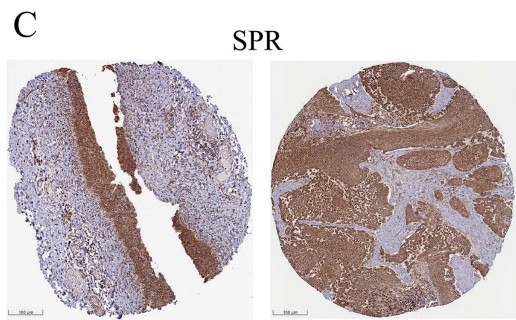

D

WBP11

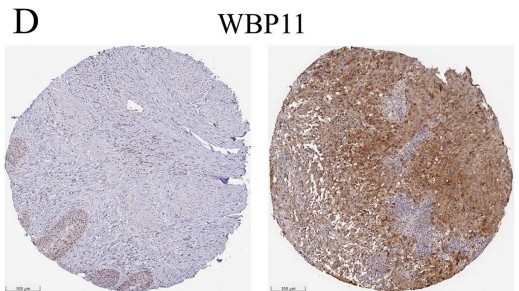

**Fig 13. Immunohistochemical staining of key genes in bladder cancer tissues.** Representative IHC staining of **(A)** ENO1, **(B)** GTF2F1, **(C)** SPR, and **(D)** WBP11 in adjacent non-tumor tissues (left) and tumor tissues (right). Scale bars are included.

concentrated. Taken together, the expression of these four key genes in bladder cancer tissues was higher than that in adjacent tissues, and their localization and expression intensity provide support at the histological level for subsequent functional studies and clinical translation.

## Molecular docking

Molecular docking analysis showed that all four screened lactylation-related candidate proteins could form stable docking conformations with BPA (Fig 14A–D). Among them, SPR showed the lowest binding energy (−7.3 kcal/mol), followed by ENO1 (−6.3 kcal/mol), whereas GTF2F1 and WBP11 both showed binding energies of −5.8 kcal/mol (Fig 14A–D), indicating relative differences in predicted binding affinity among the four candidate proteins. The binding energies of GTF2F1 and WBP11 were both −5.8 kcal/mol, indicating that they also have some affinity for BPA. A variety of interaction types, including hydrogen bonding, hydrophobic interaction, and π-π stacking, were visible in the docking model, which were mainly distributed in the active pocket region of the protein, supporting the structural plausibility of protein–ligand interactions. These results indicate that BPA can form stable binding conformations with the candidate proteins at the structural level; however, such interactions do not imply biological activity or causal relevance to bladder cancer pathogenesis.

## Molecular dynamics simulation

To further validate the conformational stability and kinetic behavior of BPA upon binding to the four key target proteins (ENO1, GTF2F1, SPR, and WBP11), we evaluated multiple key metrics of each complex system based on 100 ns molecular dynamics simulation trajectories (Fig 15A-D). First, the root mean square displacement (RMSD) analysis showed that, except for the GTF2F1-BPA system, which showed slight fluctuations at the late stage, the complexes as a whole were maintained in a reasonable stability interval, suggesting that BPA and the target proteins could form a structurally stable complex. Radius of gyration (Rg) analysis suggested that none of the four systems showed marked expansion or contraction during the simulation. In particular, the Rg curves of the ENO1–BPA and SPR–BPA systems were relatively smooth, implying that the overall protein conformations remained compact. Solvent-accessible surface area (SASA) analysis showed that the GTF2F1-BPA system had relatively large accessibility, which might be related to the structural characteristics of its surface binding site. The hydrogen bonding number statistics, on the other hand, showed that 1~2 hydrogen bonds could be stably maintained between SPR and ENO1 and BPA, which contributed to the enhancement of their binding affinity and conformational retention ability. Free energy topography (FEL) further revealed the energy conformation distribution of each system (Fig 16A-D), in which the SPR-BPA complex exhibited a single and deep energy basin, representing that it tended to a more stable low-energy conformation during the simulation process, whereas the FELs of WBP11 and ENO1 showed multiple energy minima, suggesting that they had multiple feasible low-energy conformational states. Residue mean square fluctuation (RMSF) analysis further indicated that the flexible regions of WBP11 and SPR were mainly distributed in the terminal ring structure, while ENO1 and GTF2F1 also showed some degree of conformational perturbation around the ligand binding site (Fig 16E-J), but the overall fluctuation did not exceed 6 Å, supporting the overall stability of the complex. The convergence and stability of the 100 ns molecular dynamics simulations were evaluated based on the root mean square deviation (RMSD) of the protein backbone and the protein–ligand complex. After an initial equilibration phase, the RMSD values reached a stable plateau and fluctuated within a narrow range during the later stages of the simulation, indicating that the systems had achieved dynamic equilibrium and that the simulations were sufficiently converged. In summary, the SPR-BPA system exhibited optimal stability and binding characteristics in multiple dimensions, suggesting that the SPR–BPA complex exhibits high structural stability during simulation, without implying functional regulation or causal involvement in bladder cancer progression.

 

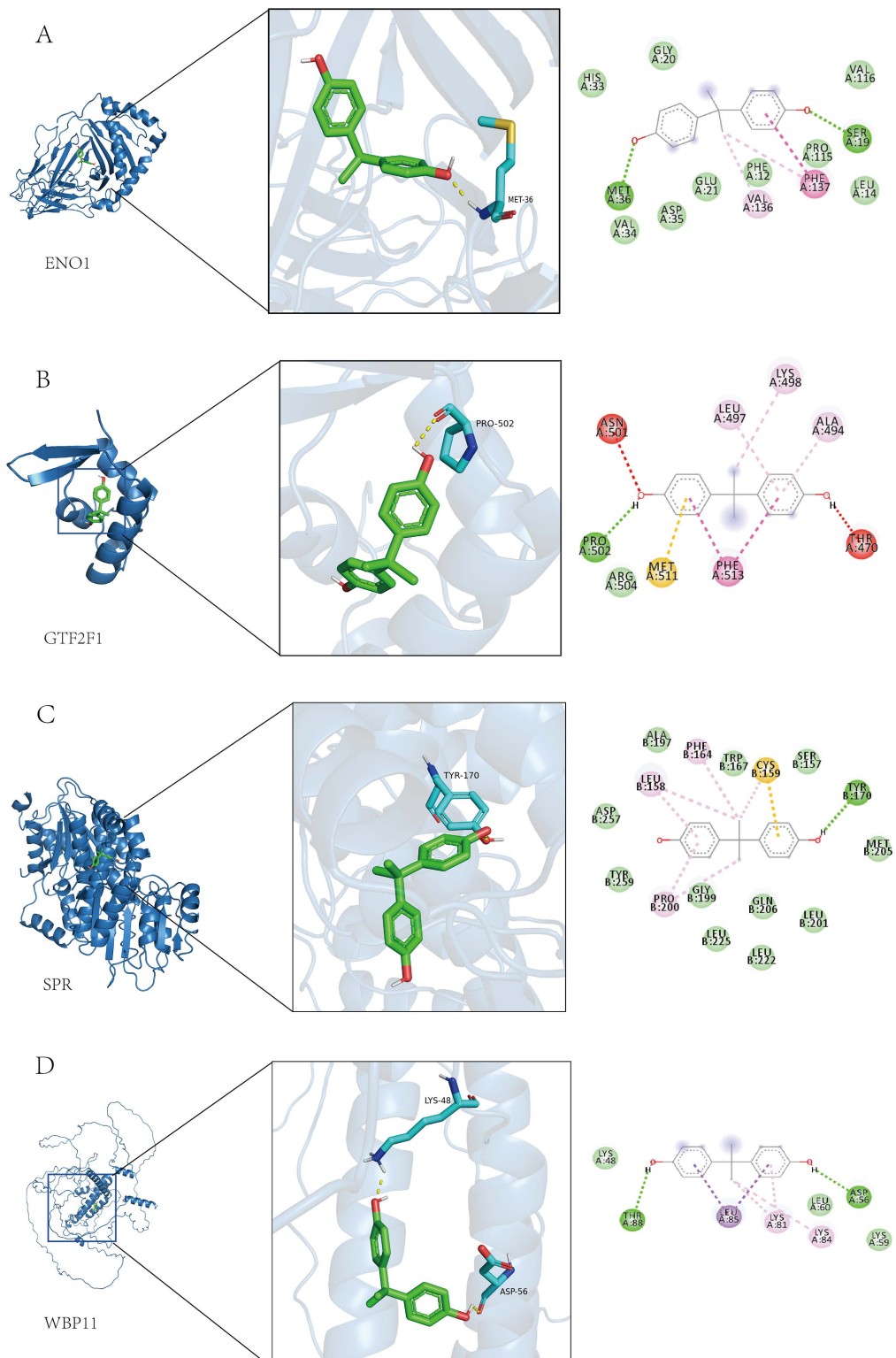

**Fig 14. Molecular docking analysis diagram. (A-D)** Docking models of BPA with ENO1, GTF2F1, SPR and WBP11, respectively, showing the overall structure of the protein, the 3D structure of the small molecule binding site, and the 2D map of the interaction from left to right. The binding energies

were −6.3, −5.8, −7.3, and −5.8 kcal/mol, respectively, indicating that all of them possessed good binding activities. To avoid misinterpretation, we indicate an empirical affinity tiering: strong (≤ −7.0 kcal/mol), moderate (−5.0 to −7.0 kcal/mol), weak (> −5.0 kcal/mol). The −5.8 kcal/mol observed for GTF2F1/WBP11–BPA represents moderate affinity, with residue-specific hydrophobic contacts and intermittent H-bonds highlighted in the 2D interaction maps, supporting biological plausibility.

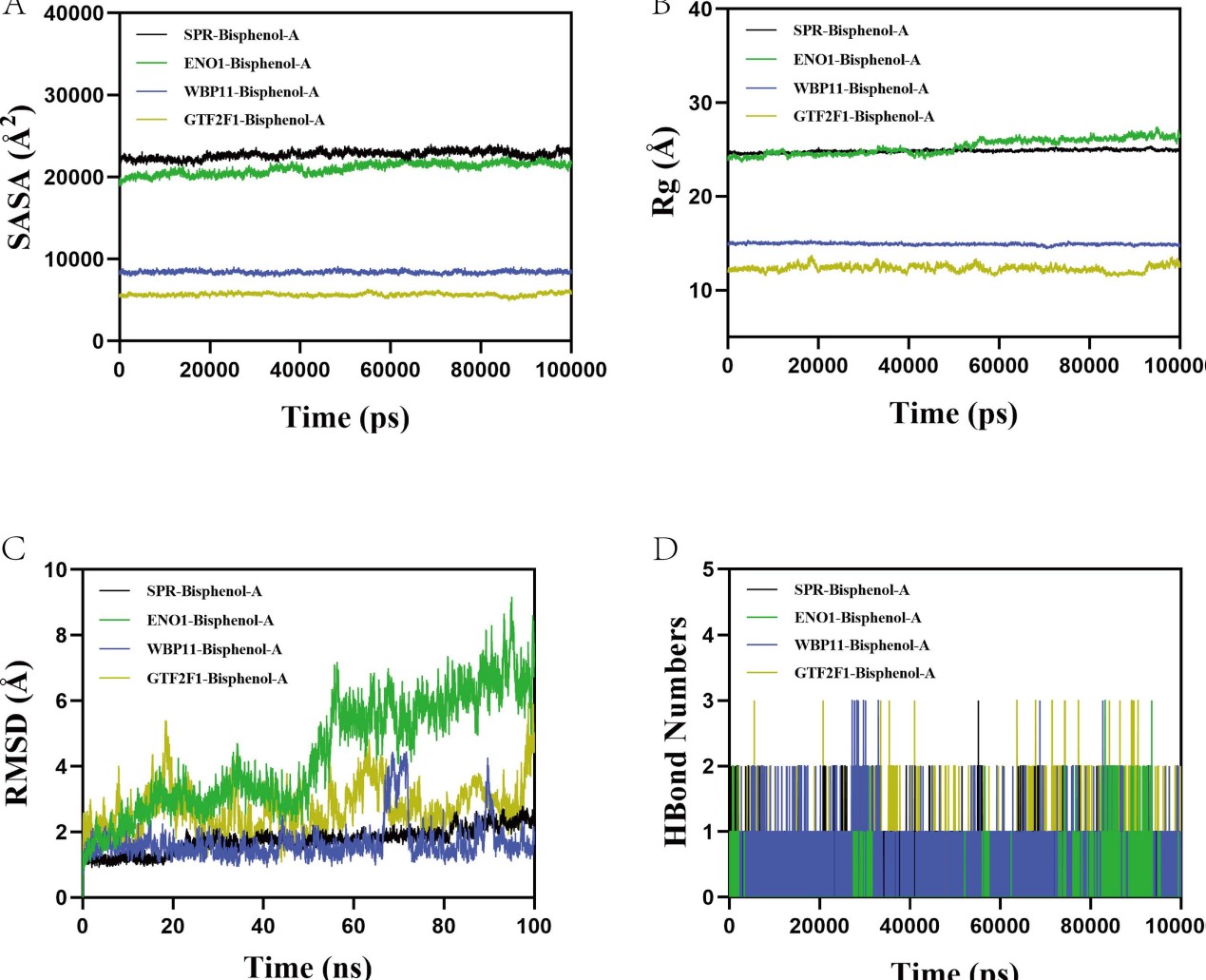

**Fig 15. Molecular dynamics simulation of protein-ligand complexes. A.** RMSD values of protein-ligand complexes over time. **B.** Rg values of protein-ligand complexes over time. **C.** SASA values of protein-ligand complexes over time. **D.** HBonds values of protein-ligand complexes over time.

## Immunofluorescence staining

Immunofluorescence staining showed that ENO1 was primarily localized in the cytoplasm of bladder epithelial cells, with markedly stronger green fluorescence signals in tumor tissues compared with their paired normal counterparts (Fig 17). Quantitative analysis demonstrated that the fluorescence intensity of ENO1 was significantly elevated in bladder cancer tissues ($P < 0.01$), indicating an increased protein expression level in malignant cells (Fig 18B).

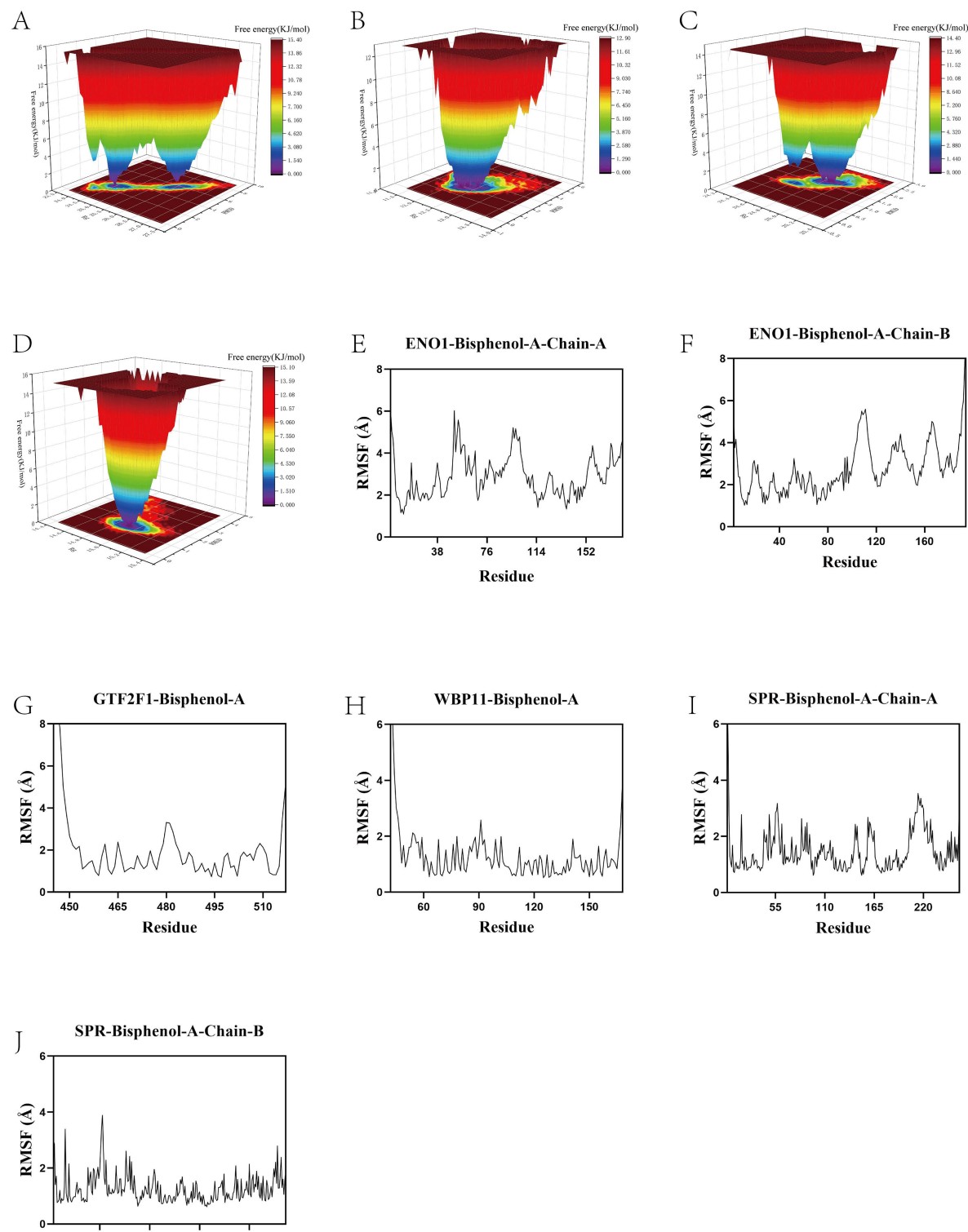

**Fig 16. (A-D) Combined with free energy landscape topography. (G-J)** RMSF values of amino acid backbone atoms of protein-ligand complexes over time.

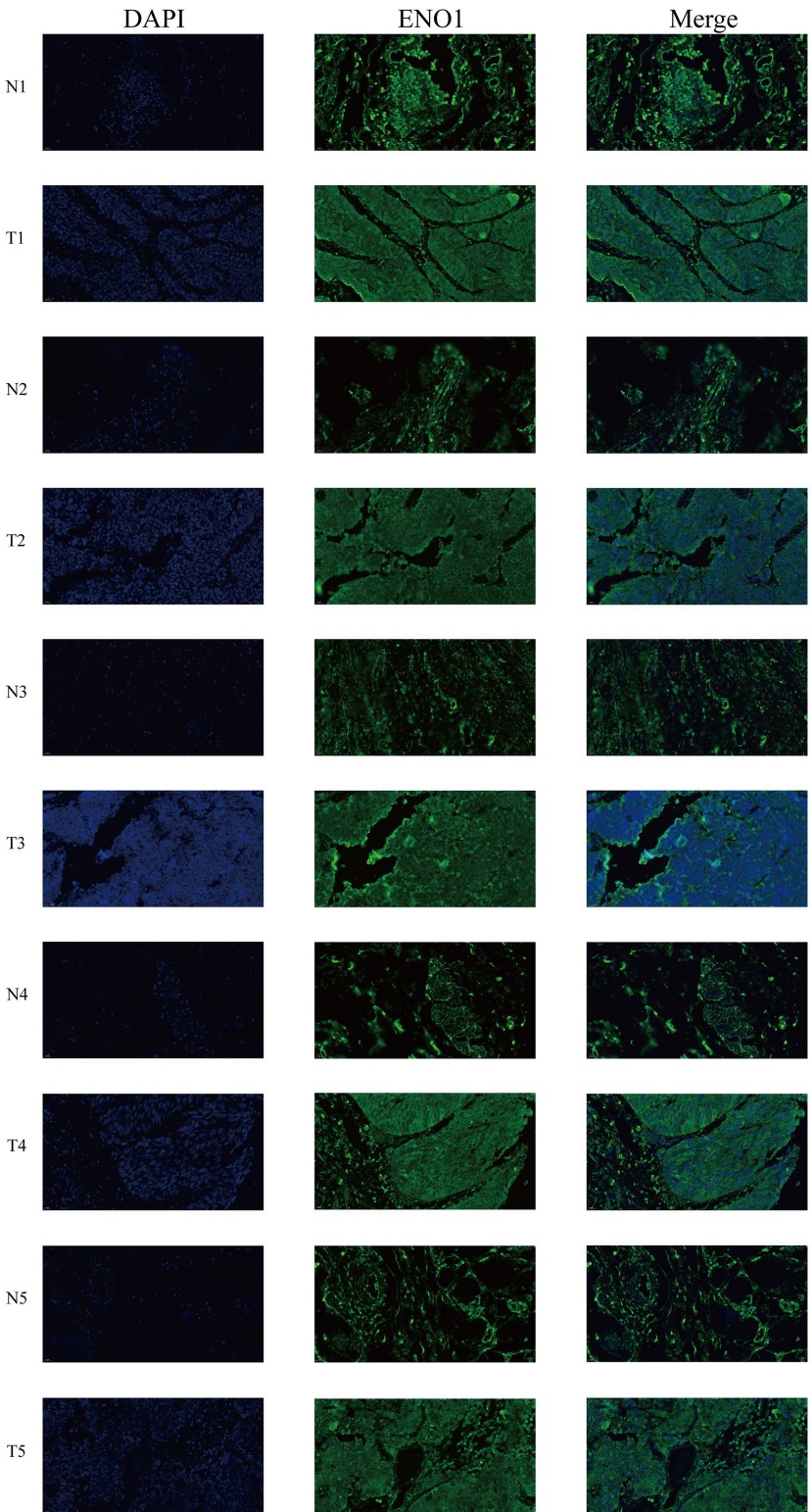

**Fig 17. Immunofluorescence staining of ENO1 in bladder cancer and normal tissues.** Representative immunofluorescence images showing ENO1 (green) and nuclei stained with DAPI (blue) in five paired tumor (T1–T5) and adjacent normal (N1–N5) bladder tissues. The merged images demonstrate enhanced ENO1 fluorescence in tumor tissues compared with normal tissues. Scale bar=20 μm.

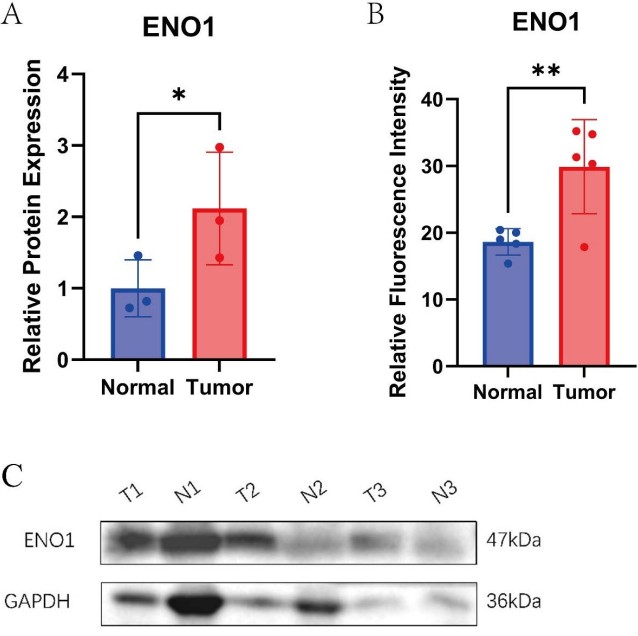

**Fig 18. Validation of ENO1 protein expression in bladder cancer tissues.** Western blot and immunofluorescence quantification results showing significantly increased ENO1 expression in bladder cancer compared with paired normal tissues. *P<0.05; **P<0.01.

## Expression validation

To further validate the expression patterns of the prioritized candidate genes in clinical tissues, we performed qPCR analysis for WBP11, SPR, and GTF2F1 in paired bladder cancer and adjacent normal tissues, and Western blot analysis for ENO1. qPCR results showed that WBP11, SPR, and GTF2F1 exhibited differential expression between bladder cancer tissues and matched adjacent tissues, providing experimental support at the transcriptional level for their prioritization as candidate genes (S2 Fig). Consistent with the immunofluorescence results, Western blot analysis further validated that ENO1 protein expression was markedly upregulated in bladder cancer tissues compared with adjacent normal tissues (Fig 18A–C, P<0.05).

Taken together, these findings provide preliminary expression-level validation for all four prioritized genes, with WBP11, SPR, and GTF2F1 supported at the mRNA level and ENO1 additionally supported at the protein level. These results strengthen the rationale for prioritizing these genes for further mechanistic investigation in bladder cancer. It should be noted that the qPCR and Western blot assays in this study were used to assess expression levels only and do not provide direct evidence for lysine lactylation modifications or BPA–protein interactions.

## Conclusion

In conclusion, this study integrated toxicogenomics data, lactylation-related gene signatures, and bioinformatics analyses to prioritize candidate genes associated with BPA in bladder cancer. Four candidate genes were identified, among which ENO1 showed the most robust diagnostic and prognostic relevance supported by experimental validation. Overall, this work presents a toxicogenomics-informed analytical framework that provides new insights into the potential involvement of environmental toxicants in bladder cancer.

## Discussion

This study focused on the potential toxicological relevance of bisphenol A (BPA) in bladder cancer and integrated WGCNA, toxicogenomics databases, and lactylation-related genes to prioritize candidate targets potentially associated with BPA-related signatures. Through machine learning, multi-omics evaluation (transcriptomic, single-cell, and protein-level data), and structural modeling, four candidate genes—ENO1, WBP11, GTF2F1, and SPR—were prioritized. Our analyses suggest that these BPA-informed candidate genes are associated with metabolic- and immune-related features in bladder cancer, providing a toxicogenomics-informed perspective linking environmental toxicant–related signatures with epigenetic-associated pathways. Notably, experimental validation in this study was performed only for ENO1; the remaining genes were prioritized through integrative computational analyses and should be interpreted as candidate targets rather than experimentally confirmed drivers. Based on these analyses, we propose a preliminary framework linking environmental toxicant–related signatures, epigenetic-associated pathways, and tumor progression [40]. Importantly, this study is exploratory in nature and aims to prioritize candidate genes and pathways rather than establish causal or mechanistic relationships.

Among the candidate genes, ENO1 emerged as the most consistently associated gene with diagnostic and prognostic outcomes in this study, suggesting potential relevance in an exploratory context. As a key glycolytic enzyme, ENO1 is involved in lactate metabolism and has been reported to be functionally linked to metabolic pathways related to lactylation in previous studies. This functional relevance likely explains its consistently elevated expression across independent cohorts and its significant association with poor prognosis in bladder cancer. By contrast, SPR, GTF2F1, and WBP11 exhibited differential expression but lacked reproducible prognostic significance, which may reflect tumor subtype heterogeneity, microenvironmental variation, or context-dependent signaling. These findings suggest that ENO1 represents a prioritized candidate for further investigation rather than an immediately clinically actionable biomarker.

BPA was initially recognized as a weak environmental estrogen, but recent studies have shown that BPA may be similar to or even stronger than estradiol in stimulating certain cellular responses [41]. In addition, BPA can act by activating the membrane receptor GPER (GPR30) and other receptors such as estrogen-related receptors (ERRs) [42]. The endocrine disrupting effects of BPA are not limited to estrogen receptor activation. It has been found that BPA and its analogs can disrupt the endocrine system through a variety of mechanisms including binding to nuclear receptors and altering signaling pathways [43]. For example, BPA can affect downstream signaling pathways by binding to estrogen receptors α (ERα) and β (ERβ) and may lead to epigenetic changes [43]. In addition, metabolites of BPA may also have high estrogenic activity, which further increases its complexity as an endocrine disruptor [44]. The estrogen-like activity of BPA is more pronounced compared to other endocrine disruptors such as phthalates. Phthalates have been found to have weak estrogen-like activity in some studies, but they act mainly by interfering with the steroidogenic pathway [45]. BPA, on the other hand, can directly affect estrogen receptor activity through multiple pathways [46]. In addition, the endocrine disrupting effects of BPA may be related to its selective binding at different receptors. Studies have shown that BPA and its analogs exhibit different binding energies and conformations when bound to ERα and ERβ, which may lead to different biological effects [46]. In the present study, we identified a set of prioritized candidates at the intersection of toxicogenomics databases, lactylation-related genes, and WGCNA modules, providing a new perspective on the association between BPA-related signatures and bladder cancer. Although BPA has been implicated in carcinogenesis in several experimental and epidemiological contexts, its relevance to bladder cancer remains incompletely established. Therefore, the present findings should be interpreted as association-based and hypothesis-generating rather than as evidence of a direct causal relationship. Consistent with this interpretation, the intersecting genes, despite showing enrichment beyond random expectation, should be regarded as statistically prioritized candidates rather than definitive biological drivers. Accordingly, the downstream core gene selection represents a prioritization step aimed at highlighting potentially relevant targets, rather than a conclusive identification of causal genes.

Among the four key lactylation-related genes screened, ENO1 was the most consistently and critically expressed marker. ENO1 (α-enolase) is a multifunctional protein involved in glycolytic processes and has been found to be overexpressed in a variety of cancers. Its role in cancer is not limited to metabolic regulation, but also involves immune escape and remodeling of the tumor microenvironment. In bladder cancer (BLCA), lactylation of ENO1 may be closely associated with metabolic reprogramming, which is one of the key factors for adaptive survival and rapid proliferation of cancer cells [47]. During metabolic reprogramming, ENO1 promotes cancer cell survival and proliferation in a hypoxic environment by regulating the glycolytic pathway. In addition, ENO1 may promote immune escape from tumors by affecting the tumor immune microenvironment (TIME) and inhibiting the infiltration of immune cells such as CD8+T cells [47]. This immune escape mechanism is particularly important in bladder cancer because it may lead to tumor resistance to immunotherapy. Studies have shown that high expression of ENO1 is associated with poor tumor prognosis, suggesting that it may serve as a potential biomarker and therapeutic target for bladder cancer. By targeting ENO1, it may help to restore the anti-tumor immune response and improve the efficacy of immunotherapy [48]. Therefore, an in-depth study of the mechanism of action of ENO1 in bladder cancer is important for the development of new therapeutic strategies. In this study, ENO1 was highly expressed in bladder cancer tissues with high diagnostic efficacy (AUC > 0.85), and high expression was significantly associated with poorer prognosis. In addition, single-cell analysis showed that ENO1 was mainly distributed in tumor epithelial cells and CD8+T cells, suggesting potential involvement in both metabolic regulation and immune-related processes in bladder cancer. Molecular docking results showed that ENO1 could form a stable complex with BPA (binding energy −6.3 kcal/mol), providing structural-level evidence of interaction.

WBP11, also known as WW domain binding protein 11, has been identified as a significant player in the progression of various cancers, including bladder cancer (BLCA). As a core splicing factor, WBP11 is involved in the regulation of alternative splicing events that are crucial for cancer cell proliferation and metastasis. In the context of ovarian cancer, WBP11 has been shown to be highly expressed and associated with poor prognosis. Its inhibition leads to a significant reduction in cancer cell proliferation and mobility, highlighting its potential as a therapeutic target [49]. In gastric cancer, WBP11 interacts with other proteins such as KIAA1199 and PTP4A3, playing a role in the epithelial-mesenchymal transition (EMT), a process that is critical for cancer metastasis. The miR-29c-KIAA1199 axis, which involves WBP11, regulates gastric cancer migration by modulating key signaling pathways like FGFR4/Wnt/β-catenin and EGFR. This interaction underscores the multifaceted role of WBP11 in cancer biology, as it not only influences splicing but also participates in signaling pathways that drive cancer progression [50]. Given these observations in other tumor types, WBP11 may also be relevant to bladder cancer biology, although its specific role in bladder cancer remains insufficiently defined. The exploration of WBP11#39;s role in BLCA could reveal new therapeutic targets and strategies, potentially improving patient outcomes by targeting the molecular pathways in which WBP11 is involved. In addition, WBP11 has a binding energy of −5.8 kcal/mol to BPA, supporting its role as a potential toxicant binding protein.

The role of GTF2F1 in bladder cancer may be closely related to immune and inflammatory responses. The immune microenvironment of bladder cancer largely influences tumor progression and response to therapy. Studies have shown that bladder cancer cells can regulate the tumor microenvironment by secreting pro-inflammatory cytokines, thereby promoting tumor migration and invasion [51]. In addition, immune-infiltrating cells in bladder cancer, such as macrophages and monocytes, play an important role in the tumor microenvironment and influence the efficacy of immunotherapy [52]. Docking analysis suggested that GTF2F1 could adopt a stable binding conformation with BPA (−5.8 kcal/mol), raising the possibility that it may be relevant to BPA-associated transcriptional features; however, this remains speculative and requires experimental validation.

Sepiapterin reductase (SPR) plays an important role in a variety of biological processes, particularly in immunity, inflammation and cancer. It has been shown that SPR plays a role in the immune response by regulating the synthesis of tetrahydrobiopterin (BH4), which in turn affects NO production [53]. In cancer research, SPR has been found to be associated with the progression of a variety of tumors.SPR affects the growth and survival of tumor cells by regulating

cell proliferation and apoptosis signaling pathways. For example, in hepatocellular carcinoma, SPR promotes cancer cell proliferation and inhibits apoptosis through the FoxO3a/Bim signaling pathway [54]. In addition, SPR has been found to be associated with breast cancer cell proliferation, and its knockdown can inhibit breast cancer cell proliferation by inducing ROS-mediated apoptosis [55]. In addition, the role of SPR in colon cancer has been investigated. By regulating the level of BH4, SPR can affect the occurrence and development of colon cancer. Studies have shown that the inhibition of SPR can reduce the number of tumors in colon cancer, which provides a new idea for the treatment of colon cancer [56]. In this study, it showed continuous up-regulation in bladder cancer tissues with significant expression at protein level and clear localization in fibroblasts and endothelial cells. Combined with molecular docking analysis, it showed the lowest binding energy (−7.3 kcal/mol) with BPA, showing a strong binding affinity at the structural level, suggesting that SPR may represent a potential BPA-interacting protein that warrants further experimental investigation.

A key contribution of this study is the systematic exploration of associations between BPA-related gene signatures and lactylation-related pathways in bladder cancer within a toxicogenomics-informed analytical framework. By integrating transcriptomic, single-cell, immunohistochemical, and molecular docking analyses with multiple machine learning approaches, this study strengthened the prioritization of candidate targets and provides a basis for subsequent experimental investigation.

At different levels of analysis, we observed certain inconsistencies in gene expression patterns. For example, GTF2F1 and SPR showed opposite expression trends between TCGA and GEO cohorts, displayed low expression in single-cell datasets, yet demonstrated moderate protein expression in immunohistochemistry. Several factors may underlie these discrepancies. First, cross-cohort comparisons are inevitably influenced by technical variation. Although we applied Com-Bat batch-effect correction, the divergent expression patterns of SPR and GTF2F1 persisted, suggesting that they cannot be fully explained by platform effects. Second, bladder cancer is characterized by marked molecular and clinical heterogeneity; the proportion of muscle-invasive versus non–muscle-invasive cases differs between TCGA and GEO, which may contribute to divergent gene expression profiles. In addition, differences in tumor microenvironment composition may also play a role: SPR is predominantly localized in fibroblasts and endothelial cells, which are underrepresented in certain datasets, thereby weakening the overall transcriptional signal. For GTF2F1, transcriptional activity is relatively limited in some cell populations, but protein expression may be stabilized through post-transcriptional regulation, explaining its more evident staining pattern. Taken together, these findings suggest that the expression of SPR and GTF2F1 is subject to both cohort-dependent variability and biological complexity, while also reflecting the complementary nature of transcriptomic and proteomic analyses.

## Limitations

Several limitations of this study should be acknowledged. First, patient-level BPA exposure data were not available; therefore, the relevance of BPA to bladder cancer in this work was inferred from curated toxicogenomics databases rather than cohort-specific exposure measurements. Second, this study did not include direct measurements of lactate metabolism, metabolic flux, or lysine lactylation marks, nor were experimental systems employed to test whether BPA directly induces lactylation changes. Accordingly, lactylation-related findings should be interpreted as functional associations inferred from transcriptomic and pathway enrichment analyses rather than direct mechanistic evidence.

Third, molecular docking and molecular dynamics simulations provided only structural-level insights and cannot establish biological activity, intracellular target engagement, or causal mechanisms without orthogonal experimental validation. Fourth, immune infiltration was estimated using computational deconvolution approaches (CIBERSORT and ssGSEA); although multiple methods were applied to improve robustness, these estimates remain indirect and lack experimental confirmation such as immunohistochemistry or flow cytometry.

Finally, the initial list of BPA-related targets retrieved from public databases was relatively broad, and despite subsequent multi-step bioinformatic filtering and experimental validation, future studies could adopt more stringent target

selection criteria. In addition, the potential synergistic effects of BPA with other environmental toxicants were not addressed and warrant further investigation

## Supporting information

**S1 Fig. Survival analyses of key genes across multiple independent cohorts.** Kaplan–Meier survival curves for WBP11, SPR, and GTF2F1 in external validation cohorts.
(TIF)

**S2 Fig. qPCR validation of key genes in paired bladder cancer and adjacent normal tissues.** Relative mRNA expression levels of SPR, WBP11, and GTF2F1 were assessed by quantitative real-time PCR.
(TIF)

**S3 Fig. Raw western blot images.** Full, uncropped western blot membranes used for ENO1 protein validation.
(PDF)

## Author contributions

**Data curation:** Hao Wang.

**Formal analysis:** Hongquan Liu.

**Funding acquisition:** Jitao Wu.

**Validation:** Jitao Wu.

**Writing – original draft:** Fengze Sun.

**Writing – review & editing:** Fengze Sun.

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
