## [Decision Letter · Decision Letter 0]

3 Feb 2026

PONE-D-25-67310Integrating Multi-Omics and Machine Learning to Decipher the Molecular Pathways of Bisphenol A-Associated Lactylation-Related Genes Driving Bladder CancerPLOS One

Dear Dr. Wu,

Thank you for submitting your manuscript to PLOS ONE. After careful consideration, we feel that it has merit but does not fully meet PLOS ONE’s publication criteria as it currently stands. Therefore, we invite you to submit a revised version of the manuscript that addresses the points raised during the review process. Please submit your revised manuscript by Mar 20 2026 11:59PM. If you will need more time than this to complete your revisions, please reply to this message or contact the journal office at plosone@plos.org. . Please include the following items when submitting your revised manuscript:

If applicable, we recommend that you deposit your laboratory protocols in protocols.io to enhance the reproducibility of your results. Protocols.io assigns your protocol its own identifier (DOI) so that it can be cited independently in the future. For instructions see: https://journals.plos.org/plosone/s/submission-guidelines#loc-laboratory-protocols. Additionally, PLOS ONE offers an option for publishing peer-reviewed Lab Protocol articles, which describe protocols hosted on protocols.io. Read more information on sharing protocols at . Additionally, PLOS ONE offers an option for publishing peer-reviewed Lab Protocol articles, which describe protocols hosted on protocols.io. Read more information on sharing protocols at https://plos.org/protocols?utm_medium=editorial-email&utm_source=authorletters&utm_campaign=protocols..

We look forward to receiving your revised manuscript.

Kind regards,

Rajesh Kumar Pathak, Ph.D.

Academic Editor

PLOS One

**Journal Requirements:**

1. When submitting your revision, we need you to address these additional requirements. Please ensure that your manuscript meets PLOS ONE's style requirements, including those for file naming. The PLOS ONE style templates can be found at https://journals.plos.org/plosone/s/file?id=wjVg/PLOSOne_formatting_sample_main_body.pdf and https://journals.plos.org/plosone/s/file?id=ba62/PLOSOne_formatting_sample_title_authors_affiliations.pdf 2. Please note that PLOS One has specific guidelines on code sharing for submissions in which author-generated code underpins the findings in the manuscript. In these cases, all author-generated code must be made available without restrictions upon publication of the work. Please review our guidelines at https://journals.plos.org/plosone/s/materials-and-software-sharing#loc-sharing-code and ensure that your code is shared in a way that follows best practice and facilitates reproducibility and reuse. 3. Thank you for stating in your Funding Statement: This work was supported by the National Natural Science Foundation of China (No. 82370690), Natural Science Foundation of Shandong Province (No. ZR2023MH241), basic research project of Yantai Science and Technology Innovation development plan (No.2023JCYJ069) and Shandong Health Science Innovation Team Building Project.  Please provide an amended statement that declares *all* the funding or sources of support (whether external or internal to your organization) received during this study, as detailed online in our guide for authors at http://journals.plos.org/plosone/s/submit-now. Please also include the statement “There was no additional external funding received for this study.” in your updated Funding Statement. Please include your amended Funding Statement within your cover letter. We will change the online submission form on your behalf. 4. Thank you for stating the following financial disclosure: This work was supported by the National Natural Science Foundation of China (No. 82370690), Natural Science Foundation of Shandong Province (No. ZR2023MH241), basic research project of Yantai Science and Technology Innovation development plan (No.2023JCYJ069) and Shandong Health Science Innovation Team Building Project.   Please state what role the funders took in the study.  If the funders had no role, please state: "The funders had no role in study design, data collection and analysis, decision to publish, or preparation of the manuscript." If this statement is not correct you must amend it as needed. Please include this amended Role of Funder statement in your cover letter; we will change the online submission form on your behalf. 5. Please note that your Data Availability Statement is currently missing the DOI/accession number of each dataset and a direct link to access each database. If your manuscript is accepted for publication, you will be asked to provide these details on a very short timeline. We therefore suggest that you provide this information now, though we will not hold up the peer review process if you are unable. 6. Please upload a new copy of Figures 1 – 16, as the detail is not clear. Please follow the link for more information:  https://journals.plos.org/plosone/s/figures 7. PLOS ONE now requires that authors provide the original uncropped and unadjusted images underlying all blot or gel results reported in a submission’s figures or Supporting Information files. This policy and the journal’s other requirements for blot/gel reporting and figure preparation are described in detail at https://journals.plos.org/plosone/s/figures#loc-blot-and-gel-reporting-requirements and https://journals.plos.org/plosone/s/figures#loc-preparing-figures-from-image-files. When you submit your revised manuscript, please ensure that your figures adhere fully to these guidelines and provide the original underlying images for all blot or gel data reported in your submission. See the following link for instructions on providing the original image data: https://journals.plos.org/plosone/s/figures#loc-original-images-for-blots-and-gels.  In your cover letter, please note whether your blot/gel image data are in Supporting Information or posted at a public data repository, provide the repository URL if relevant, and provide specific details as to which raw blot/gel images, if any, are not available. Email us at plosone@plos.org if you have any questions. 8. If the reviewer comments include a recommendation to cite specific previously published works, please review and evaluate these publications to determine whether they are relevant and should be cited. There is no requirement to cite these works unless the editor has indicated otherwise.

**Additional Editor Comments:**

The manuscript presents interesting findings. However, it requires major revision to address significant shortcomings in transparency, methodology and figure quality.

Reviewers' comments:

Reviewer's Responses to Questions

**Comments to the Author**

1. Is the manuscript technically sound, and do the data support the conclusions?

Reviewer #1: Yes

Reviewer #2: Yes

Reviewer #3: Partly

2. Has the statistical analysis been performed appropriately and rigorously? 

Reviewer #1: Yes

Reviewer #2: Yes

Reviewer #3: No

3. Have the authors made all data underlying the findings in their manuscript fully available?

Reviewer #1: Yes

Reviewer #2: Yes

Reviewer #3: Yes

4. Is the manuscript presented in an intelligible fashion and written in standard English?

Reviewer #1: Yes

Reviewer #2: Yes

Reviewer #3: No

5. Review Comments to the Author

**Reviewer #1:** The authors present an integrative bioinformatics study combining public transcriptomic datasets, toxicogenomics annotations, immune deconvolution, single-cell resources, and molecular docking/MD to propose putative BPA-associated, lactylation-related drivers in bladder cancer. The topic is of potential interest; however, several major conceptual and methodological issues currently limit the strength of the central claims particularly the implied linkage between BPA exposure, lactylation biology, and bladder cancer pathogenesis. In its current form, the manuscript reads as a hypothesis-generating analysis rather than a biologically supported BPA–lactylation–cancer study.The authors present an integrative bioinformatics study combining public transcriptomic datasets, toxicogenomics annotations, immune deconvolution, single-cell resources, and molecular docking/MD to propose putative BPA-associated, lactylation-related drivers in bladder cancer. The topic is of potential interest; however, several major conceptual and methodological issues currently limit the strength of the central claims particularly the implied linkage between BPA exposure, lactylation biology, and bladder cancer pathogenesis. In its current form, the manuscript reads as a hypothesis-generating analysis rather than a biologically supported BPA–lactylation–cancer study.

This study does not include patient-level BPA exposure measurements linked to the transcriptomic specimens. Consequently, the manuscript’s repeated “BPA-associated” framing is not evidence-based and appears to rest almost entirely on curated database annotations rather than cohort-specific data. In its current form, the work should be reframed as toxicogenomics-informed candidate prioritization. Alternatively, the authors must add direct exposure-linked analyses to substantiate any BPA–tumor association claims.

Combining a large toxicogenomics-derived BPA target list with WGCNA modules and a literature-derived lactylation set is highly prone to producing seemingly "plausible" overlaps even under null conditions, particularly given gene-set size effects and network/module selection flexibility. The manuscript currently provides no rigorous statistical calibration. Without these analyses, the alleged specificity of the 74-gene intersection—and by extension the downstream “core gene” selection—remains unsubstantiated and may reflect methodological artifacts rather than biology.

While enrichment analyses may be consistent with a lactylation-related storyline, the manuscript offers no direct evidence of altered lactate metabolism/flux, no measurement of lysine lactylation marks, and no demonstration that BPA induces lactylation changes in bladder cancer-relevant systems. As written, the mechanistic interpretation is not warranted. The authors must either substantially temper the mechanistic claims or provide direct validation.

Molecular docking and MD simulations can at best suggest structural plausibility of binding. They cannot establish biological activity, intracellular target engagement, downstream pathway modulation, or causal relevance to tumor phenotypes. Accordingly, any strong mechanistic claims derived primarily from docking/MD are not justified and should be removed or reframed as speculative, unless supported by orthogonal experimental evidence.

The manuscript refers to multiple “key genes,” yet experimental validation is provided only for ENO1. This selective validation weakens the claim that all identified candidates constitute robust key drivers/biomarkers.

Figure 2: The resolution of the figures is too low to clearly read labels and interpret the content. Please provide higher-resolution versions of all figures in accordance with the journal’s image quality requirements.

Line 636: The manuscript attributes the near-perfect training AUCs mainly to overfitting. Please elaborate on the likely causes of this overfitting and clarify what steps were taken to prevent information leakage and ensure unbiased performance estimates.

Additional comment

Line 47: Please define abbreviations at first mention by providing the full term followed by the abbreviation in parentheses.

Line 67: The wording “studies” appears inconsistent with the citations provided (only one reference is cited). Please revise for accuracy.

Line 79: Please spell out the full names of GEO and TCGA at first mention.

Line 482: Please check and standardize spacing in P-value reporting

Line 530: Please review and correct spacing/formatting issues.

Lines 732 and 737: Please check and correct spacing between words/sentences for readability.

Please review the manuscript to ensure consistent reference formatting throughout.

A comprehensive check is needed to ensure consistent abbreviation definitions and usage across the manuscript.

**Reviewer #2:** Major comments Major comments

1. The author used reliable sources from bioinformatics studies.

2. The aim of the study has not been clearly stated.

3. The clarity of the figures should be improved.

4. I appreciate the authors for incorporating the wet-lab results

5. The protein–protein interaction study reported 74 genes, clarify whether the four identified genes are included among them. If so, highlight it.

6. Explain clearly the evidence that supports the convergence and reliability of the 100 ns MD simulation in the study?

7. I encourage authors, the manuscript would benefit from extensive editing and proper formatting.

8. I suggest author to revise the manuscript to improve readability. The content is valuable and provides important insights for researchers.

9. I encourage author to provide a clarity in the conclusion to highlight the key outcomes and novelty of the study.

Decision: The manuscript has been accepted after the reviewers’ comments were addressed

**Reviewer #3:** The manuscript is technically sound and includes the necessary data. However, the writing has gaps, and a few parts of the statistical analyses need clearer explanation. Addressing these points will improve the clarity and overall quality of the work. Other detailed comments are attached.The manuscript is technically sound and includes the necessary data. However, the writing has gaps, and a few parts of the statistical analyses need clearer explanation. Addressing these points will improve the clarity and overall quality of the work. Other detailed comments are attached.

6. PLOS authors have the option to publish the peer review history of their article (what does this mean?). If published, this will include your full peer review and any attached files.). If published, this will include your full peer review and any attached files.

.

Reviewer #1: No

Reviewer #2: No

Reviewer #3: No

---

## [Author Response · Author response to Decision Letter 1]

5 Feb 2026

Dear Editor,

First of all, thank you very much for taking time out of your busy schedules to read and revise our manuscript. Thank you for your valuable comments. You have thoroughly advised us on the structure, content, research methodology and results of our paper. It has played a very important role in improving the quality of our paper, and on behalf of my co-authors, I would like to thank you for giving us the opportunity to revise and improve the quality of our article.

We have carefully read the reviewers' comments and your comments and have made changes, which are highlighted in yellow in the paper. We have tried our best to revise our manuscript based on the comments. and also fulfill the requirements of the journal. Attached is the revised version which we would like to submit for your consideration. We have written a point-by-point response letter for this purpose, the details of which are at the end of the letter.

In response to the journal's specific requirements, we have fully addressed each item: we have adjusted the manuscript to comply with PLOS ONE's style and file naming guidelines as per the provided templates; ensured all author-generated code is shared in line with best practices for reproducibility; updated the Funding Statement to include all sources of support and the required statement "There was no additional external funding received for this study"; clarified the funders' role in the study as specified; supplemented the Data Availability Statement with DOIs, accession numbers, and direct database links; re-uploaded high-clarity versions of Figures 1–18 following the journal's figure preparation guidelines; provided the original uncropped and unadjusted images for all blot/gel results as required.

Reviewer #1

1.The authors present an integrative bioinformatics study combining public transcriptomic datasets, toxicogenomics annotations, immune deconvolution, single-cell resources, and molecular docking/MD to propose putative BPA-associated, lactylation-related drivers in bladder cancer. The topic is of potential interest; however, several major conceptual and methodological issues currently limit the strength of the central claims particularly the implied linkage between BPA exposure, lactylation biology, and bladder cancer pathogenesis. In its current form, the manuscript reads as a hypothesis-generating analysis rather than a biologically supported BPA–lactylation–cancer study.

Response:

We thank the reviewer for this important clarification. We agree that the present study is primarily hypothesis-generating in nature. Accordingly, we have revised the manuscript throughout to explicitly frame our work as a toxicogenomics-informed and exploratory analysis aimed at prioritizing candidate genes and pathways, rather than establishing biologically validated causal mechanisms. Overly strong mechanistic or causal language has been systematically tempered, particularly in the Abstract, Discussion, and Conclusion.

The revisions are in lines 29-32, 941-945, 1062-1089 of the revised manuscript

2.This study does not include patient-level BPA exposure measurements linked to the transcriptomic specimens. Consequently, the manuscript’s repeated “BPA-associated” framing is not evidence-based and appears to rest almost entirely on curated database annotations rather than cohort-specific data. In its current form, the work should be reframed as toxicogenomics-informed candidate prioritization. Alternatively, the authors must add direct exposure-linked analyses to substantiate any BPA–tumor association claims.

Response:

We fully acknowledge this limitation. This study does not include patient-level BPA exposure measurements, and all BPA-related inferences are derived from curated toxicogenomics annotations in CTD. To address this concern, we revised the terminology throughout the manuscript (e.g., replacing “BPA-associated” with “BPA-linked candidates”) and explicitly clarified this limitation in the Methods, Discussion, and Limitations sections.

The revisions are in lines 117-118, 893-904, 941-949 of the revised manuscript

3. Combining a large toxicogenomics-derived BPA target list with WGCNA modules and a literature-derived lactylation set is highly prone to producing seemingly "plausible" overlaps even under null conditions, particularly given gene-set size effects and network/module selection flexibility. The manuscript currently provides no rigorous statistical calibration. Without these analyses, the alleged specificity of the 74-gene intersection—and by extension the downstream “core gene” selection—remains unsubstantiated and may reflect methodological artifacts rather than biology.

Response:

We thank the reviewer for this important methodological concern. We agree that intersecting large gene sets may produce seemingly plausible overlaps under null conditions, particularly due to gene-set size effects and module selection variability. To address this issue, we have added statistical calibration analyses in which random gene sets matched for size were repeatedly sampled to evaluate the expected distribution of intersection sizes under null conditions.

These analyses demonstrated that the observed 74-gene overlap exceeded random expectation and was not solely driven by methodological artifacts. In addition, we have revised the manuscript to clarify that the intersecting genes and the downstream core gene selection represent statistically prioritized candidates rather than definitive biological drivers, and we have appropriately tempered the related interpretations in the Discussion.

The revisions are in lines 162-169, 486-488, 1062-1089 of the revised manuscript

4. While enrichment analyses may be consistent with a lactylation-related storyline, the manuscript offers no direct evidence of altered lactate metabolism/flux, no measurement of lysine lactylation marks, and no demonstration that BPA induces lactylation changes in bladder cancer-relevant systems. As written, the mechanistic interpretation is not warranted. The authors must either substantially temper the mechanistic claims or provide direct validation.

Response:

We thank the reviewer for this important and constructive comment. We agree that the original version of the manuscript may have overinterpreted enrichment-based results in a mechanistic manner. As correctly noted, the present study does not include direct measurements of lactate metabolism or metabolic flux, does not assess lysine lactylation marks, and does not experimentally demonstrate BPA-induced lactylation changes in bladder cancer–relevant systems.

In response, we have substantially revised the manuscript to temper all mechanistic interpretations related to lactylation. Specifically, we clarified the definition of “lactylation-related genes” as genes functionally associated with lactylation-related pathways rather than proteins confirmed to be lactylated, revised the Results section to describe enrichment and association patterns without causal or mechanistic language, and explicitly acknowledged the absence of direct lactylation and metabolic validation in the Discussion and Limitations sections.

These revisions ensure that the lactylation-related findings are presented as transcriptomics- and enrichment-based functional associations within a toxicogenomics-informed framework, rather than as direct evidence of altered lactylation states or mechanisms.

The revisions are in lines 566-570, 582-585 of the revised manuscript

5. Molecular docking and MD simulations can at best suggest structural plausibility of binding. They cannot establish biological activity, intracellular target engagement, downstream pathway modulation, or causal relevance to tumor phenotypes. Accordingly, any strong mechanistic claims derived primarily from docking/MD are not justified and should be removed or reframed as speculative, unless supported by orthogonal experimental evidence.

Response:

We thank the reviewer for this important and constructive comment. We fully agree that molecular docking and molecular dynamics simulations can only provide evidence at the structural level and cannot establish biological activity, intracellular target engagement, downstream pathway modulation, or causal relevance to tumor phenotypes.

In response to this concern, we have carefully revised the manuscript to remove or substantially temper any mechanistic claims derived primarily from docking or MD analyses. The Results section now presents docking and MD findings strictly as evidence of structural plausibility and interaction stability, without implying functional or causal effects. In addition, we have revised the Discussion to explicitly clarify the evidentiary boundaries of these computational approaches and emphasized that any functional implications of BPA–protein interactions remain speculative and require independent experimental validation.

Furthermore, the limitations of molecular docking and molecular dynamics simulations are now explicitly acknowledged in the Limitations section. These revisions ensure that the interpretation of docking/MD results is fully aligned with their appropriate scope and level of evidence.

The revisions are in lines 778-782, 822-831, 974-976, 1027-1026 of the revised manuscript

5. The manuscript refers to multiple “key genes,” yet experimental validation is provided only for ENO1. This selective validation weakens the claim that all identified candidates constitute robust key drivers/biomarkers.

Response:

We thank the reviewer for this comment. We agree that experimental validation was performed only for ENO1. Accordingly, we have revised the Discussion section to explicitly distinguish ENO1 as the only gene supported by experimental validation, while the remaining genes are described as computationally prioritized candidates that warrant further experimental investigation. This revision ensures that the interpretation of “key genes” accurately reflects the different levels of supporting evidence.

The revisions are in lines 893-900 of the revised manuscript

6.Figure 2: The resolution of the figures is too low to clearly read labels and interpret the content. Please provide higher-resolution versions of all figures in accordance with the journal’s image quality requirements.

Response:

We thank the reviewer for this comment. Figure 2 has been regenerated and uploaded separately as a high-resolution version in accordance with the journal’s image quality requirements. The updated figure provides improved label clarity and overall readability.

7. Line 636: The manuscript attributes the near-perfect training AUCs mainly to overfitting. Please elaborate on the likely causes of this overfitting and clarify what steps were taken to prevent information leakage and ensure unbiased performance estimates.

Response:

We thank the reviewer for this important comment. The near-perfect training AUCs are primarily attributable to the high dimensionality of the transcriptomic features relative to the sample size, combined with the strong correlation structure among genes, which can lead to overly optimistic performance on the training data even when regularized models are used.

To mitigate overfitting and prevent information leakage, several precautions were taken. First, all feature selection and model training procedures were performed exclusively within the training set, and the validation datasets were kept completely independent and were not used at any stage of feature selection or model optimization. Second, cross-validation was applied within the training cohort to tune model parameters, and performance was assessed using held-out validation cohorts to obtain unbiased estimates. Third, consistent preprocessing steps were applied separately to training and validation datasets to avoid data leakage.

Accordingly, model performance was interpreted primarily based on validation and external cohort results rather than training AUCs. We have clarified these points in the revised manuscript to better explain the source of overfitting and the measures taken to ensure unbiased performance evaluation.

The revisions are in lines 268-272, 279-281,665-661 of the revised manuscript

Additional comment

Line 47: Please define abbreviations at first mention by providing the full term followed by the abbreviation in parentheses.

Line 67: The wording “studies” appears inconsistent with the citations provided (only one reference is cited). Please revise for accuracy.

Line 79: Please spell out the full names of GEO and TCGA at first mention.

Line 482: Please check and standardize spacing in P-value reporting

Line 530: Please review and correct spacing/formatting issues.

Lines 732 and 737: Please check and correct spacing between words/sentences for readability.

Please review the manuscript to ensure consistent reference formatting throughout.

A comprehensive check is needed to ensure consistent abbreviation definitions and usage across the manuscript.

Response:

We thank the reviewer for these helpful comments. The manuscript has been carefully revised accordingly. Abbreviations are now defined at their first mention, and the full names of GEO and TCGA are provided upon initial appearance. The wording at Line 67 has been corrected to reflect the appropriate number of cited references. Spacing and formatting issues, including P-value reporting and inter-word spacing at the indicated lines, have been corrected and standardized throughout the manuscript. In addition, we conducted a comprehensive review to ensure consistent abbreviation usage and reference formatting across the entire manuscript.

The revisions are highlighted in yellow in the text

Reviewer #2

1.The author used reliable sources from bioinformatics studies.

Response:

We thank the reviewer for this positive comment. We appreciate the recognition of the reliability of the bioinformatics data sources used in this study.

2. The aim of the study has not been clearly stated.

Response:

We thank the reviewer for this helpful comment. We agree that the study aim was not stated with sufficient clarity. In the revised manuscript, we have explicitly stated the aim of the study in the Introduction, clarifying that our objective was to systematically identify and prioritize potential BPA-associated, lactylation-related candidate genes in bladder cancer using integrative bioinformatics and multi-level validation approaches.

The revisions are in lines 76-78 of the revised manuscript

3. The clarity of the figures should be improved.

Response:

We thank the reviewer for this comment. All figures have been regenerated and uploaded as separate high-resolution files to improve clarity and readability. The revised figures now meet the journal’s image quality standards.

4. I appreciate the authors for incorporating the wet-lab results

Response:

We thank the reviewer for this positive comment. We appreciate the reviewer’s recognition of the inclusion of wet-lab experimental validation to support the bioinformatics findings.

5. The protein–protein interaction study reported 74 genes, clarify whether the four identified genes are included among them. If so, highlight it.

Response:

We thank the reviewer for this comment. Yes, all four prioritized candidate genes are included among the 74 genes in the protein–protein interaction network. These genes have been highlighted directly in the PPI figure to improve clarity and facilitate interpretation.

6. Explain clearly the evidence that supports the convergence and reliability of the 100 ns MD simulation in the study?

Response:

We thank the reviewer for this important comment. The convergence and reliability of the 100 ns molecular dynamics simulations were evaluated using standard stability metrics, primarily the root mean square deviation (RMSD) of the protein backbone and protein–ligand complexes. Following an initial equilibration period, the RMSD values reached a stable plateau and fluctuated within a narrow range during the latter portion of the simulations, indicating that the systems had achieved dynamic equilibrium.

In addition, complementary indicators such as RMSF and radius of gyration profile

---

## [Decision Letter · Decision Letter 1]

12 Mar 2026

PONE-D-25-67310R1Integrating Multi-Omics and Machine Learning to Decipher the Molecular Pathways of Bisphenol A-Associated Lactylation-Related Genes Driving Bladder CancerPLOS One

Dear Dr. Wu,

Thank you for submitting your manuscript to PLOS ONE. After careful consideration, we feel that it has merit but does not fully meet PLOS ONE’s publication criteria as it currently stands. Therefore, we invite you to submit a revised version of the manuscript that addresses the points raised during the review process. Please submit your revised manuscript by Apr 26 2026 11:59PM. If you will need more time than this to complete your revisions, please reply to this message or contact the journal office at plosone@plos.org. . Please include the following items when submitting your revised manuscript:

If applicable, we recommend that you deposit your laboratory protocols in protocols.io to enhance the reproducibility of your results. Protocols.io assigns your protocol its own identifier (DOI) so that it can be cited independently in the future. For instructions see: https://journals.plos.org/plosone/s/submission-guidelines#loc-laboratory-protocols. Additionally, PLOS ONE offers an option for publishing peer-reviewed Lab Protocol articles, which describe protocols hosted on protocols.io. Read more information on sharing protocols at . Additionally, PLOS ONE offers an option for publishing peer-reviewed Lab Protocol articles, which describe protocols hosted on protocols.io. Read more information on sharing protocols at https://plos.org/protocols?utm_medium=editorial-email&utm_source=authorletters&utm_campaign=protocols..

We look forward to receiving your revised manuscript.

Kind regards,

Rajesh Kumar Pathak, Ph.D.

Academic Editor

PLOS One

Journal Requirements:

Reviewers' comments:

Reviewer's Responses to Questions

**Comments to the Author**

1. If the authors have adequately addressed your comments raised in a previous round of review and you feel that this manuscript is now acceptable for publication, you may indicate that here to bypass the “Comments to the Author” section, enter your conflict of interest statement in the “Confidential to Editor” section, and submit your "Accept" recommendation.

Reviewer #1: (No Response)

Reviewer #2: All comments have been addressed

Reviewer #3: All comments have been addressed

2. Is the manuscript technically sound, and do the data support the conclusions?

Reviewer #1: Yes

Reviewer #2: Yes

Reviewer #3: Yes

3. Has the statistical analysis been performed appropriately and rigorously? 

Reviewer #1: Yes

Reviewer #2: Yes

Reviewer #3: Yes

4. Have the authors made all data underlying the findings in their manuscript fully available?

Reviewer #1: Yes

Reviewer #2: Yes

Reviewer #3: Yes

5. Is the manuscript presented in an intelligible fashion and written in standard English?

Reviewer #1: Yes

Reviewer #2: Yes

Reviewer #3: Yes

6. Review Comments to the Author

Reviewer #1: The authors have adequately addressed my major concerns. The revised manuscript is now more appropriately framed as a hypothesis-generating, toxicogenomics-informed prioritization study, and the conclusions are commensurate with the level of evidence. I have no further major concerns.

Reviewer #2: The author has addressed the requested questions.

Major comments:

1. The figures are still blurry and unclear. Please improve the clarity

2. I would encourage authors the alignment of the article needs to improve.

3. I suggest the authors explore ways to make the manuscript more reader-friendly, if feasible.

4. ENO1 has a well-known candidate in bladder cancer, the associations of WBP11, GTF2F1, and SPR with this disease have not been clearly established and should be justified by the author.

Reviewer #3: (No Response)

7. PLOS authors have the option to publish the peer review history of their article (what does this mean?). If published, this will include your full peer review and any attached files.). If published, this will include your full peer review and any attached files.

.

Reviewer #1: No

Reviewer #2: No

Reviewer #3: No

---

## [Author Response · Author response to Decision Letter 2]

14 Mar 2026

Dear Dr. Pathak and Reviewers,

Thank you very much for your careful evaluation of our manuscript and for the constructive comments and suggestions. We sincerely appreciate the time and effort you and the reviewers devoted to assessing our work. These comments have been helpful in improving the clarity, rigor, and overall quality of the manuscript.

We have carefully considered all comments from the Academic Editor and reviewers and have revised the manuscript accordingly. In the revised version, all changes have been highlighted in yellow for ease of review, and a separate marked-up file with track changes has also been provided, in accordance with the journal’s requirements. In addition, we have prepared a detailed point-by-point response to each comment below.

During revision, we substantially improved the manuscript at both the scientific and presentation levels. Briefly, we refined the overall structure and readability of the text, clarified the interpretation of the study as a toxicogenomics-informed, hypothesis-generating prioritization analysis, improved figure quality by re-exporting and re-uploading all figures as high-resolution files, and added new supplementary analyses and experimental validation to strengthen the justification for the prioritized candidate genes. Specifically, we included additional survival analyses for WBP11, GTF2F1, and SPR across multiple independent cohorts (Supplementary Figure 1) and added tissue-level qPCR validation for these three genes (Supplementary Figure 2). We also revised the Discussion to more clearly distinguish the stronger evidence supporting ENO1 from the more exploratory status of WBP11, GTF2F1, and SPR.

We have also addressed the journal’s technical and reporting requirements in the revised submission, including revision of the manuscript files and figure files in accordance with PLOS ONE guidelines, updating the relevant reporting statements where needed, and providing the required supporting materials.

Reviewer

1. The figures are still blurry and unclear. Please improve the clarity

Response:

Thank you for your valuable suggestion. We have thoroughly improved the quality of the figures in the revised manuscript. All figures in the main text, together with the newly added supplementary figures included in this revision, were reassembled and exported as high-resolution images at 600 dpi to enhance sharpness, readability, and overall presentation quality. In addition, all revised figure files were uploaded separately in high-resolution format to ensure that image quality is preserved during review. We hope that these revisions have adequately addressed the reviewer’s concern.

2. I would encourage authors the alignment of the article needs to improve.

Response:

Thank you for this helpful comment. We have revised the manuscript to improve its overall alignment, coherence, and consistency. In the revised version, we streamlined the wording and structure of the Abstract, Introduction, Results, and Discussion so that the study is presented more consistently as a toxicogenomics-informed, hypothesis-generating prioritization analysis rather than a mechanistic or causal study. We also improved the correspondence between the figure order and the presentation of results in the main text, refined transitions between sections, and clarified the relative interpretation of the four prioritized genes. In particular, ENO1 is now discussed as the gene with the strongest experimental support in this study, whereas WBP11, GTF2F1, and SPR are more clearly described as prioritized candidate targets supported mainly by integrative analyses and supplementary expression validation. We believe these revisions have substantially improved the alignment of the manuscript.

3. I suggest the authors explore ways to make the manuscript more reader-friendly, if feasible.

Response:

Thank you for this valuable suggestion. We have revised the manuscript to improve its readability at both the language and structural levels. In the revised version, we simplified several long or dense sentences, improved transitions between paragraphs and sections, and refined the presentation of key findings to make the overall narrative clearer and easier to follow. In addition, the manuscript was further polished with the assistance of a professional scientific editing service. For ease of review, all major readability-related revisions have been highlighted in yellow in the revised manuscript. We hope that these changes have improved the reader-friendliness of the article.

The revisions are highlighted in yellow in the revised manuscript.

4. ENO1 has a well-known candidate in bladder cancer, the associations of WBP11, GTF2F1, and SPR with this disease have not been clearly established and should be justified by the author.

Response:

Thank you for this important comment. We agree that, compared with ENO1, the associations of WBP11, GTF2F1, and SPR with bladder cancer are less well established in the current literature. In the revised manuscript, we have therefore clarified that these three genes should be interpreted as prioritized candidate genes identified through an integrative, toxicogenomics-informed screening framework, rather than as definitively established bladder cancer drivers.

Specifically, their inclusion was based on a multi-step prioritization strategy integrating WGCNA, BPA-related toxicogenomics targets, lactylation-related gene sets, and machine learning–based feature selection, through which WBP11, GTF2F1, and SPR were consistently retained together with ENO1. To further strengthen the justification for these three genes, we additionally performed survival analyses across multiple independent datasets. In particular, WBP11 was further evaluated in GSE154261, IMvigor210, and GSE69795; SPR was evaluated in GSE31684, GSE13507, and IMvigor210; and GTF2F1 was evaluated in GSE13507, GSE19423, GSE31684, GSE39281, GSE69795, and IMvigor210. These additional prognostic analyses have been added as Supplementary Figure S1.

In addition, we supplemented the study with tissue-level qPCR validation for WBP11, GTF2F1, and SPR, which provided preliminary experimental support at the transcriptional level for their inclusion as candidate genes. These qPCR results have been added as Supplementary Figure S2.

We hope that these additional analyses and textual revisions have adequately addressed the reviewer’s concern.

Supplementary Figure S1

Supplementary Figure S2

Thank you again for your time. We hope that the revised manuscript is now suitable for publication in PLOS ONE.

Sincerely,

Jitao Wu

on behalf of all authors

---

## [Decision Letter · Decision Letter 2]

30 Mar 2026

Integrating Multi-Omics and Machine Learning to Decipher the Molecular Pathways of Bisphenol A-Associated Lactylation-Related Genes Driving Bladder Cancer

PONE-D-25-67310R2

Dear Dr. Wu,

We’re pleased to inform you that your manuscript has been judged scientifically suitable for publication and will be formally accepted for publication once it meets all outstanding technical requirements.

An invoice will be generated when your article is formally accepted. Please note, if your institution has a publishing partnership with PLOS and your article meets the relevant criteria, all or part of your publication costs will be covered. Please make sure your user information is up-to-date by logging into Editorial Manager at Editorial Manager® and clicking the ‘Update My Information' link at the top of the page. For questions related to billing, please contact  and clicking the ‘Update My Information' link at the top of the page. For questions related to billing, please contact billing support..

Kind regards,

Rajesh Kumar Pathak, Ph.D.

Academic Editor

PLOS One

Additional Editor Comments (optional):

The manuscript is suitable for publication.

Reviewers' comments:

Reviewer's Responses to Questions

**Comments to the Author**

1. If the authors have adequately addressed your comments raised in a previous round of review and you feel that this manuscript is now acceptable for publication, you may indicate that here to bypass the “Comments to the Author” section, enter your conflict of interest statement in the “Confidential to Editor” section, and submit your "Accept" recommendation.

Reviewer #2: All comments have been addressed

2. Is the manuscript technically sound, and do the data support the conclusions?

Reviewer #2: Yes

3. Has the statistical analysis been performed appropriately and rigorously? 

Reviewer #2: Yes

4. Have the authors made all data underlying the findings in their manuscript fully available?

Reviewer #2: Yes

5. Is the manuscript presented in an intelligible fashion and written in standard English?

Reviewer #2: Yes

6. Review Comments to the Author

Reviewer #2: 1. The author has made the manuscript more reader friendly.

2. I encourage authors to carefully review the manuscript formatting and alignment.

3. I encourage authors to improve the image quality, as some plots appear blurry. Example Figure1 the workflow and other example Figure 5 heatmap.

7. PLOS authors have the option to publish the peer review history of their article (what does this mean?). If published, this will include your full peer review and any attached files.). If published, this will include your full peer review and any attached files.

.

Reviewer #2: No

---

## [Editor Report · Acceptance letter]

PONE-D-25-67310R2

PLOS One

Dear Dr. Wu,

I'm pleased to inform you that your manuscript has been deemed suitable for publication in PLOS One. Congratulations! Your manuscript is now being handed over to our production team.

Kind regards,

on behalf of

Dr. Rajesh Kumar Pathak

Academic Editor

PLOS One